# The Lawn as a Social and Cultural Phenomenon in Perth, Western Australia

Maria Ignatieva [1,*], Michael Hughes [2], Ashok Kumar Chaudhary [3] and Fahimeh Mofrad [1]

1   School of Design, The University of Western Australia, 35 Stirling Highway, Perth, WA 6009, Australia; fahimeh.mofrad@uwa.edu.au
2   School of Environmental and Conservation Sciences, Murdoch University (MU), Murdoch, WA 6150, Australia; m.hughes@murdoch.edu.au
3   School of Agriculture and Environment, The University of Western Australia, 35 Stirling Highway, Perth, WA 6009, Australia; ashok.chaudhary@uwa.edu.au
*   Correspondence: maria.ignatieva@uwa.edu.au

**Abstract:** Lawns, introduced in Australia through English colonial heritage, dominate public spaces in cities, serving various ecosystem functions. Australian lawns consist of non-native grasses that differ from native original vegetation and require intensive management and maintenance. This study explores public perspectives on urban lawns in Perth, Western Australia, an area largely overlooked in ecological and social research in the context of Australia compared to Europe and North America. This paper presents empirical research on public perceptions of urban lawns and alternatives in Perth, Western Australia. The study explores social values and preferences regarding traditional lawns and new options, considering visual appearance, uses, and maintenance. Findings from an online questionnaire, involving 171 respondents, identified seven categories based on a content analysis of lawn definitions: flat area; ground covered by grass; maintained; non-native vegetation; open space; recreational space; and turf grass. The results revealed that respondents most value lawns for aesthetics, cooling and recreation (exercises, walking pets, as a transit area, passive recreation, and social gatherings). At the same time, participants demonstrated an environmental awareness of lawns and the necessity of revisiting the existing planning and maintenance routine based on irrigation and intensive mowing by considering several alternative solutions. While valuing new solutions such as Scaevola patches in dedicated areas and "weedy lawns", participants still preferred alternatives closest in appearance to a conventional lawn (e.g., lawn grass with Dichondra and lawn grass with clover). The study emphasizes the need for a 'blended model' of urban lawns, combining durability with heat-resistant, biodiverse vegetation to address social values and environmental concerns.

**Keywords:** lawns as a social phenomenon; definitions; public views and values; purpose and use of lawns; alternative solutions; Western Australia; Perth





## 1. Introduction

The lawn is one of the most common and familiar elements of urban green spaces in cities around the globe and is an integral part of public parks and private gardens, roadsides, industrial areas and sports grounds. Lawns originated in Europe from managed meadows and pasturelands during the medieval period. From the middle of the 19th century, lawns turned into the most desirable and fundamental feature within both public and private gardens in the United Kingdom, Europe and European colonies. By the beginning of the 21st century, lawns had become synonymous with urban landscapes globally, despite differences in climatic and socio-economic conditions [1].

There are many different definitions of a "lawn", but we define a lawn as an intentionally established "plant community consisted of predominantly grass species (cultivars), which are sown by seeds or planted using vegetative parts and could contain spontaneously occurring (unwanted) herbaceous species ("lawn weeds")" [1] (p. 5) and which are the

subject of constant ongoing maintenance. Lawns serve multiple ecosystem services and are used for different recreation activities. One of the most important aesthetical features that make lawns attractive to urban citizens is the green colour and its beauty. The lawn is an important element of landscape designs and is used as a green carpet for displaying feature plants (trees, shrubs and flowerbeds) and decorative elements such as pieces of sculpture, fountains and benches [1].

One of the main distinctive characteristics of lawns is turf/sod which is a complex entity consisting of dense intertwined grass shoots above ground and a mesh of bounded living stolons and roots that are in symbiosis with soil fauna below ground. Lawns require specific establishment techniques (the preparation of soil and seed mixtures or specially prepared rolled turf in turf farms) and management regimes (mowing, herbicide application, aeration, fertilising and watering). These techniques are aimed at maintaining targeted grass species, controlling weeds and mosses, and keeping a desirable grass height to create a walkable surface.

Over the past 30 years, research has explored the ecosystem service functions provided by lawns including their aesthetical function, cooling effect, heat island effect mitigation via transpiration and evaporation, carbon sequestration, regulation of the water cycle, habitat provision, enhancing private property values, and offering social, recreation activities and even therapeutical values such as tranquillity. Furthermore, analyses have been conducted on ecological components such as biodiversity, the role of lawns in urban planning and landscape design, and people's perceptions (e.g., an analysis of socio-cultural reasons of public attachment to lawns) in European countries and the USA [2–5].

However, growing concerns have arisen about the urbanisation process that has led to the visual and biological homogenisation of urban green spaces globally. In particular, there is a growing worry about the loss of biodiversity and the increasing pressures associated with climate change. This concern has drawn attention to the issue of urban lawns and the associated requirement for intensive management and the use of significant amounts of water and energy. These issues have encouraged the development of alternative sustainable solutions to conventional lawns that require less intensive management and fewer resources. Examples of alternative solutions such as naturalistic plantings, pictorial meadows, grass-free lawns, and transformation of conventional lawns into meadows or pastures can be seen in the urban areas of the United Kingdom, Germany, Sweden, France, Spain, Portugal, and the USA [1,6–9] (Figure 1).

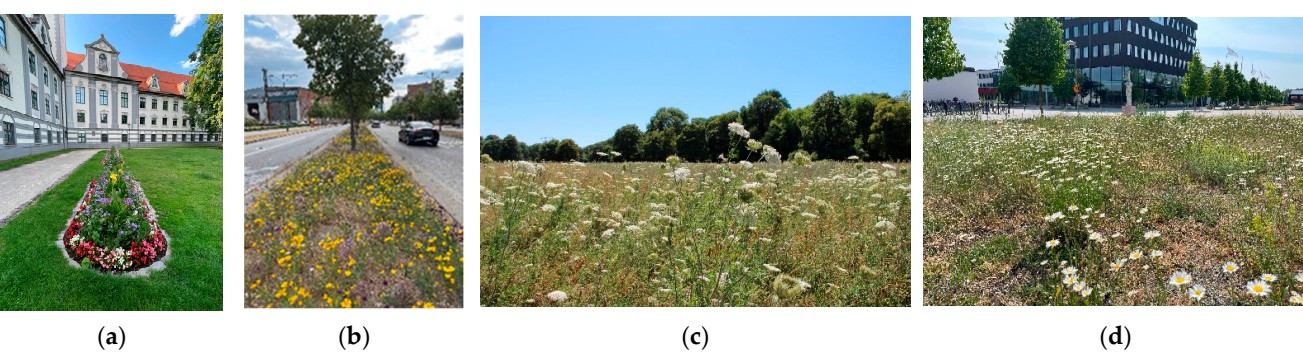

(**a**)        (**b**)        (**c**)        (**d**)

**Figure 1.** Conventional smooth green carpet in Augsburg, Germany (**a**) versus alternatives: pictorial meadow from annual plants in Malmö, Sweden (**b**), urban meadow in English Garden, Munich, Germany (**c**) and grass-free lawn in Uppsala, Sweden (**d**). Photos: M. Ignatieva.

*Lawns in Australia*

Lawns were introduced to Australia by European colonists at the end of the 18th century. Despite the popularity and wide use of lawns for 200 years, there are only a few sources on the cultural and social history of the use of lawns in Australia. Butler-Bowdon [10] discussed the development of lawns in Australian cities, from the middle-class houses' patches of front and back lawns in mixed-use gardens of the late 19th century to

minimalist manicured lawns in Australian modernist gardens of the 1920–1930s which aimed to be an exterior room of a family house and demonstrated the social norms of the society. Particularly for the first European settlers who resided in dry and hot places of Australia such as Perth, lawns were an important tool for "civilising" towns as an opposition to the unfamiliar and very alien-looking native environment, "the bush" [11]. For the successful establishment of lawns, Australian settlers had to find appropriate grass species and employ effective techniques. They used grass species from South Africa, Asia, America and Europe to create turf/sod, enabling its use for recreational and sport purposes. The introduced lawn species needed new soils, the application of fertilisers, and the use of herbicides to successfully grow. Unlike Europe, where both grass species and weeds are native and thus considered to be a part of urban nature, lawns in Australia were used to delineate "civilised" urban areas from native vegetation ("native nature") and to create spaces with short cut grass, specifically designed for recreation and sports [12] (Figure 2).

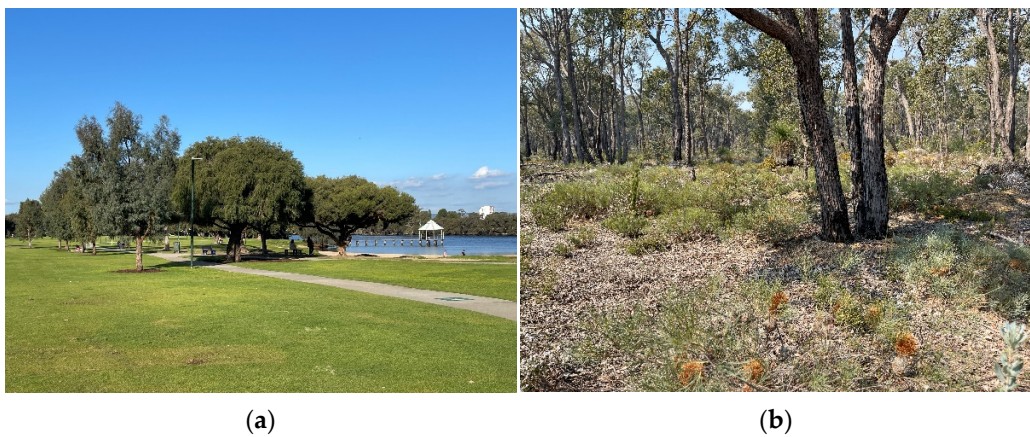

(**a**)                      (**b**)

**Figure 2.** Lawn in a public park in Perth (**a**) and remnant of native "bush" (*Eucalyptus* woodland) within Perth metropolitan area (**b**). Photos: M. Ignatieva.

By the middle of the 20th century, lawns in Australian cities dominated public open spaces (POS) including parks and recreational reserves, foreshores, public gardens, nature reserves, civic areas and promenades. Thus, lawns in Australia are the subject of built infrastructure and designed landscapes [12]. There is a growing number of studies on the role of public open spaces and their benefits for human well-being. For instance, parks are assessed for their usage and activities [13]. Another example is a study conducted by Dickinson [14], which examined the overall non-material benefits of urban green spaces in Perth without specifying the typology of green spaces. There have been several studies on Australian lawns as an agricultural crop, e.g., how to prepare soils, grow lawn grass species and maintain a good lawn that provides a green smooth surface for play and joy [15]. While lawns in Australia face growing challenges to uphold a high-quality surface in an increasingly dry climate with a decreasing water supply, there is a lack of research dedicated exclusively to understanding lawns as an ecological and social phenomenon.

Growing urbanisation and the creation of cultivated and irrigated green spaces by replacing native forests and wetlands resulted in the dramatic loss of unique Australian ecosystems. That pushed to the development of a strong environmental movement aimed at conserving, protecting and restoring unique native biodiversity. Regarding growing climate change impacts, including a drying climate, higher temperatures, increased water scarcity and pollution due to herbicide and fertiliser use, the use of native plants and the rewilding of urban green spaces have gained more support. The main arguments for using native vegetation to replace lawns are to reduce water demand, improve the sense of local identity, enhance biodiversity and improve associated ecosystem services in urban environments. However, there are very few studies on the theoretical as well as practical application of potential alternative lawns. Grose [16] was one of the first who argued for a

rethink of the use and purpose of lawns in Perth's open spaces and argued for the necessity to establish lawns only in designated areas where grassed areas provide amenities and require less maintenance. One of the main arguments for the call to use native vegetation is that the local vegetation can reduce water demand and return native biodiversity to the urban environment. The reduction of water use is the red thread of new studies on public preferences for different landscape design scenarios that could decrease the irrigation needs in public parks. One example is redesigning park grounds and substituting some watered lawn areas with draught-tolerant native vegetation, groundcovers and mulch [17]. One particular type of lawn in Perth has received more attention as a potential for alternative solutions. These are lawns located on verges (areas between streetscapes and private property boundaries) [18].

In Australia, "true" alternative solutions to lawns (surfaces that are like lawns, e.g., grass-dominated communities that can accommodate trampling and regular mowing) could be inspired by different local grassy ecosystems and even by "hybrid" models where native herbaceous species are blended with the lawn's grasses. This approach could be similar to northern hemisphere countries such as Germany, Sweden, and France mimicking native meadows, pastureland, or open margins of temperate forests, which support grasses and low-growing vegetation [1,6,8]. Some Australian native grasses can be recommended as an alternative to classic lawn grasses, for example, *Microlaena stipoides*, *Danthonia* spp. and *Themeda triandra* [19], but they are not often used in the landscape design of urban public open spaces and need further studies.

Most existing lawn alternatives (especially in areas lacking native grassland biomes such as Western Australia) may be designed using perennial groundcovers and low-growing shrubs. The purpose of alternative solutions is to reduce the number of unused lawn surfaces, thus decreasing the use of water and avoiding ecological homogeneity by employing different landscape design patterns (colour and texture) as well as providing more biodiversity and ecologically friendly wildlife habitats [1]. However, compared to Europe, these alternatives in Australia do not provide an equivalent to conventional lawns.

Compared to Europe, lawns in Australia have never been an object of a separate scientific study, neither as a specific urban biotope or social phenomenon nor as an aesthetical element of public and private green spaces. This study aims to fill this gap in recognising lawns as complex socio-cultural and ecological entities in Australia. The main research question is as follows: "How do the people of Perth, Western Australia, define and use private and public lawns as well as understand their role in an urban environment?" Another research question is the following: "What are the current social values and preferences of different socio-economic groups of people toward existing traditional lawns and lawn alternatives that introduce new species, designs and management strategies?" One subquestion of this study is the following: "What is the current maintenance and perceived quality of local lawns in Perth?" This question aims to reveal the existing maintenance routine of urban lawns and, in perspective, suggest more sustainable and economical approaches. This particular question was included due to the recent decision to change the sprinkler roster for scheme water users in Perth and Mandurah, reducing from three days per week to two days per week [20].

This research is a part of a large interdisciplinary research project investigating the phenomenon of lawns in Perth from environmental (biodiversity), social (perceptions and attitudes), and design and planning perspectives to test alternative sustainable design solutions for urban lawns in a drying climate using empirical data and innovative technologies.

## 2. Materials and Methods

### 2.1. Study Area: Perth, Western Australia

Perth is the sprawling capital city of Western Australia founded by British colonists in 1829, appropriating the territory of the Wadjuk Noongar people. Perth is situated on a coastal plain, a narrow strip between the Darling Range and the Indian Ocean. The Wadjuk Noongar nation has occupied Perth and surrounding areas and managed the landscape for

at least 40,000 years. The city is located within what is known as the Southwest Australian Floristic Region (SWAFR), which is one of 35 biodiversity hotspots worldwide. The remnant native vegetation in the region is characterised by the richness of exceptional plant species, with about 8000 species, and high endemism [21]. Perth experiences a Mediterranean type of climate with hot dry summers (24.6 °C is the mean max temperature) and warm wet winters (12.7 °C is the mean minimum). The city has the highest level of daily sunshine among all Australian capital cities [22] which allows people to develop a year-round outdoor lifestyle. The annual rainfall is 850 mm of which about 90% occurs between April and October.

Perth's climate is different from the climates of major cities on the east coast of Australia. For example, Sydney is located in a humid subtropical climate with mild and cool winters, warm and hot summers, and no dry seasons, with an average rainfall of around 1175 mm a year. Melbourne is located in a temperate oceanic climate with an average rainfall of around 650 mm a year and even a rain distribution pattern throughout the year. The closest climate among Australian cities is Adelaide in South Australia which is also located in a Mediterranean type of climate with an average rainfall around 550 mm a year. However, Perth is hotter than Adelaide in the summer, and Adelaide is cooler in winter.

Below Perth, there are three layers of aquafers: superficial, the deeper Leederville Aquifer, and the lower Yarragadee Aquifer with ancient waters that are 40,000 years old and with an extraordinary capacity of water (1000 cubic kilometres of water). No other Australian city and not many cities around the globe have such extensive aquafers. This unique hydrological system supports Perth's rivers, wetlands, minor waterways, and diverse native vegetation. It supplies an important part of potable scheme water [23]. For many decades, Perth's green spaces, including lawns, had the privilege of being watered several times a week. However, the mean annual rainfall in Perth has declined significantly since the 1970s with a 30% reduction in stream flow. While Perth is located in a relatively wetter area of Western Australia, the reduction in rainfall and stream flow means there is a strong reliance on subterranean aquifers and the desalination of sea water to supply potable water for the growing population of Perth [24]. The mixture of groundwater (47%) and desalinated water (53%) makes Perth unique compared to other state capitals in Australia that are highly dependent upon surface water [23].

The Perth metropolitan area covers about 1640 km$^2$ and extends 150 km north–south along the Swan Coastal Plain confined between the Indian Ocean to the west and a scarp (the Darling Scarp) on the eastern boundary that restricts the east–west urban sprawl to about 50 km wide. The most recent census, conducted in 2021, indicated there were nearly 2.1 million residents living in the greater Perth area [25] (Figure 3). Perth's urban planning structure is based on the classic colonial grid that was subsequently transformed into a downtown low-density suburbia pattern in the 20th century. The city is dominated by single-story, owner-occupied homes with small gardens. Green and blue spaces within the city mainly consist of publicly accessible nature reserves based on remnants of native vegetation and designed urban public parks, including pocket parks planted with mainly exotic plants or a mixture of native and exotic species.

Lawns are a prominent feature of Perth's landscapes. In Perth's 'Urban Zone' (as defined by the Metropolitan Region Scheme or MRS), 7% is covered by lawns, 12% is made up of tree canopies and the remaining 81% is impervious or built areas (buildings, roads, pavements). Public open spaces (POS), water bodies and industrial zones are separate types of land use in the MRS. Within areas zoned public open space (including regionally significant parks and recreational reserves, foreshores, public gardens, nature reserves, civic areas and promenades), lawns cover 26% of the total area. Lawns in private gardens and verges (areas in the public road reserve between the carriageway and the boundary of property) are the leading category (53%), followed by lawns in smaller local parks (37%). The domination of lawns in public parks and gardens reflected the accepted picturesque–gardenesque landscape design "formula" where tree belts, tree groves and single trees are scattered on short-cut lawns [1].

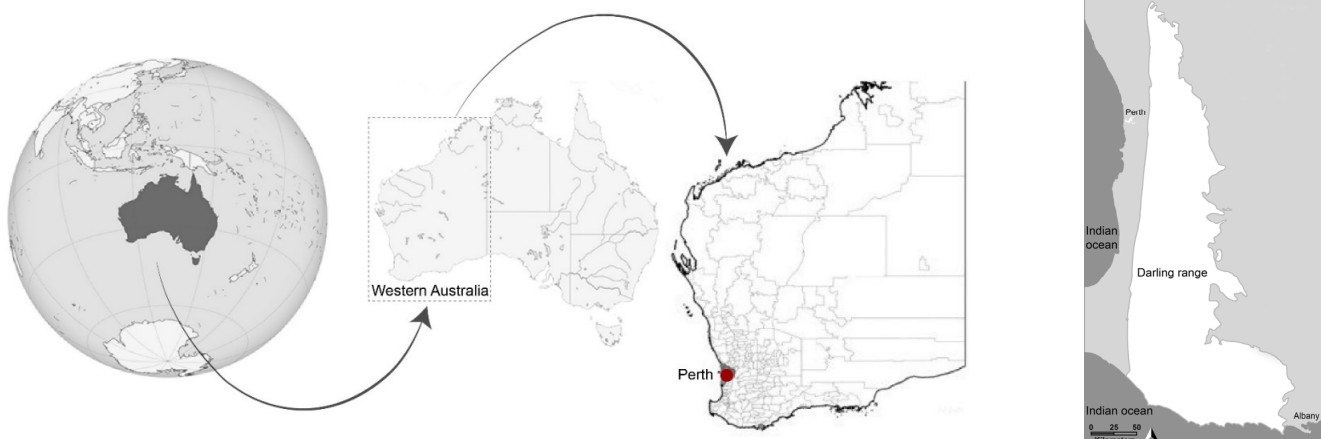

**Figure 3.** Map location of Perth, Western Australia. Map produced by authors based on the figures sourced and adapted from Wikimedia. CC BY 3.0 https://commons.wikimedia.org/wiki/File: Australia_on_the_globe_(Australia_centered).svg accessed on 7 December 2023 (**left image**) and https://creativecommons.org/licenses/by-sa/3.0/deed.en accessed on 7 December 2023 (**right image**).

*2.2. Online Questionnaire*

An online questionnaire was developed to ascertain Perth's residents' perceptions and preferences associated with lawn alternatives, using the Qualtrics online survey tool [26]. The questionnaire included a total of 37 open and multiple-choice questions about different aspects of urban lawns (Table S1) with four major themes related to the research questions:

1. General understanding of what a lawn is and its main purpose;
2. Uses of a lawn (human and non-human);
3. Maintenance and perceived quality of local lawns;
4. Perceptions and preferences associated with lawn alternatives.

The general understanding of what a lawn is, and its main purpose, was ascertained by asking respondents to write a brief definition and then a brief comment about the main purpose of urban lawns. To measure lawn and lawn alternative preferences, respondents rated and then ranked a series of five images of lawn alternatives (Figure 4). The images were selected based on an analysis of existing experimental lawn alternatives of living labs in Perth [27]. An experimental trial in Perth, "Lawn as an Ecological and Cultural Phenomenon", was designed in 2021. We also analysed existing alternatives in Europe and Australia including pictorial meadows, naturalistic plantings, grass-free lawns, weedy lawns and woody meadows. Our vision of alternative solutions is based on understanding the essence of lawns as durable surfaces that can endure a certain amount of trampling and other kinds of human activity.

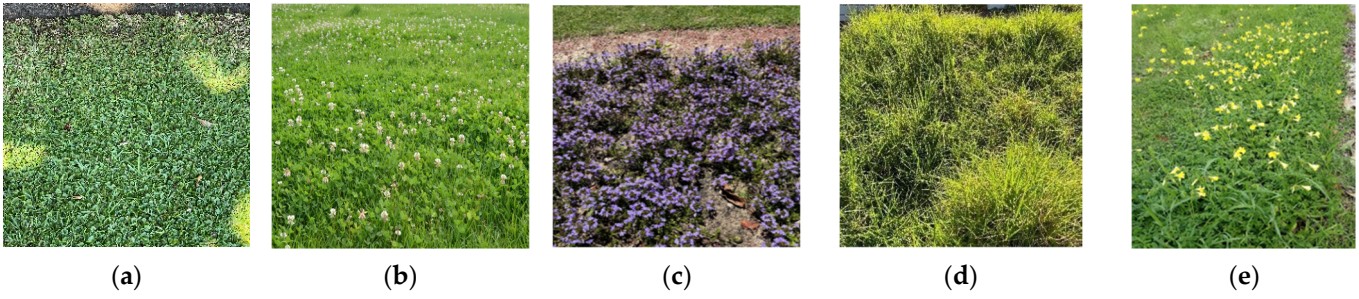

(**a**)　　　　　(**b**)　　　　　(**c**)　　　　　(**d**)　　　　　(**e**)

**Figure 4.** Lawn alternative images used in the questionnaire. Lawn with turf grasses and Dichondra (**a**); lawn with turf grasses and clover (**b**); Scaevola patches in dedicated spaces (**c**); 'old' uncut high grass lawn (**d**); biodiverse lawn with turf grasses and flowering weeds (**e**). Photos: M. Ignatieva.

While there are some limitations in using photographs to depict an image to the survey participants [28,29], photo survey remains the most commonly used and reliable methodology for the aesthetic evaluation of a landscape that includes a variety of environmental contexts including urban environments, agricultural fields, and wilderness [30–33]. The value of using images in the questionnaires for a better understanding and visualisation of ecological information in urban landscapes has also been acknowledged by American [34], English [35] and Australian [17] scholars. The most recent example of using direct photographs in existing demonstration lawn trials in a questionnaire related to residents' perceptions of and preference for the lawn alternative was conducted in the city of Xianyang, China [36].

The questions were related to private lawns as well as to lawns in the nearest public parks.

The survey was distributed online using a snowball sampling method. An invitation to complete the questionnaire was sent via email to the researchers' professional and social network residents in the Perth metropolitan area. The email invitation asked participants to forward the questionnaire to others in their own social networks. This snowball sampling approach extended the sample to a broader pool of participants. The respondent sample included participants from across the Perth metropolitan area.

*2.3. Data Analysis*

The data analysis included a content analysis, descriptive statistics and modelling to explain the effects between respondent characteristics and other survey responses. Written responses to open-ended questions ascertaining the respondents' general understanding of what a lawn is and its main purpose were analysed using inductive content analysis [37]. The responses were manually coded to identify common terms, which were grouped by similar meaning to identify key response categories. Coding was conducted independently by the researchers and then compared and discussed to resolve any discrepancies. Descriptive statistics were used to describe respondent demographics, uses and management of lawns and lawn preferences. Finally, a logit model was used to estimate the effects of explanatory variables on lawn use, management, perceptions and preferences (Table 1).

**Table 1.** Description of dependent and independent variables for logit model.

| Variable Name | Description |
| --- | --- |
| Private/or public | Dependent variable: Value 1 if respondent had private lawn, 0 otherwise |
| Age group | Value 1 if age of the respondent between 18–24 years, 2 if 25–34 years, 3 if 35–44 years, 4 if 45–54 years, 5 if 55–64 years, 6 if 65 years or above |
| Identity | Value 1 if male, 2 if female, 3 if third gender, 4 if prefer to self-describe, 5 if prefer not to say |
| Dwelling | Value 1 if detached house, 2 if flat or apartment or townhouse, 3 if others |
| Residence | Value 1 if Australian, 2 if non-Australian |
| Employment | Value 1 if full-time, 2 if part-time, 3 if home duties, 4 if unemployed, 5 if retired, 6 if student, 7 if others |
| Education | Value 1 if primary, 2 if secondary, 3 if university graduate, 4 if university postgraduate, 5 if other, 6 if vocational training |

Logit models are commonly used to examine factors influencing preferences, perceptions and behaviours such as the adoption of agroforestry practices [38–40], slash and burn agriculture [41], REDD+ [42] and composting [43]. For this study, the logit model is specified as follows:

$$logit\ (Y) = \ln\left[\frac{p_i}{1 - p_i}\right] = \alpha + \beta_1 X_{1i} + \beta_2 X_{2i} + \beta_3 X_{3i} + \cdots + \beta_k X_{ki} \tag{1}$$

where *Y* represents a private lawn (*Y* = 1 if respondents have a private lawn and *Y* = 0 if respondents have no private lawns), subscript *i* refers to the *i*-th observation in the sample, *p* is the probability of the adoption of a private lawn. *α* is the intercept term, $X_1, X_2, X_3, \ldots, X_k$ are explanatory variables and $\beta_1, \beta_2, \beta_3, \ldots, \beta_k$ are coefficients of explanatory variables. These coefficients were estimated using the maximum likelihood method.

### 2.4. Respondent Demographics

The majority of respondents were females (53%) aged between 35 and 64 years of age (55%) (Table 2). Most respondents had a university-level education (81%). They worked full-time (61%) and lived in a detached house (82%) with a private lawn, as shown in Table 2. These data generally align with the census data for the Perth region, where 51% of the population is female, the median age is 37 years, 57% work full-time and 79% live in a detached house [25]. The level of education is notably higher for our survey sample when compared with the greater Perth population (26% with a university degree). Analysis indicated that age generally did not influence the responses to the survey.

**Table 2.** Socio-demographics characteristics of respondents.

| Identity | n | % | Age Group | n | % |
|---|---|---|---|---|---|
| Male | 61 | 41.78% | 18–24 | 14 | 9.52% |
| Female | 78 | 53.42% | 25–34 | 25 | 17.01% |
| Non-binary/third gender | 1 | 0.68% | 35–44 | 38 | 25.85% |
| Prefer to self-describe | 2 | 1.37% | 45–54 | 42 | 28.57% |
| Prefer not to say | 4 | 2.74% | 55–64 | 15 | 10.20% |
| Total responses | 146 | 100% | 65 or older | 13 | 8.84% |
| Place of residence | | | Total responses | 147 | 100% |
| Australia | 144 | 97.96% | Employment type | | |
| Outside Australia | 3 | 2.04% | Working full-time | 89 | 60.54% |
| Total responses | 147 | 100% | Working part-time/casual | 25 | 17.01% |
| Type of dwelling | | | Home duties | 5 | 3.40% |
| Detached house | 120 | 82.19% | Unemployed | 0 | 0.00% |
| Flat/apartment/townhouse | 22 | 15.07% | Retired | 8 | 5.44% |
| Other (please specify) | 4 | 2.74% | Student | 20 | 13.61% |
| Total responses | 146 | 100% | Other | 0 | 0.00% |
| Education | | | Total responses | 147 | 100% |
| Primary school | 0 | 0.00% | | | |
| Secondary school (high school) | 13 | 8.90% | | | |
| University graduate | 54 | 36.99% | | | |
| University higher degree (postgraduate) | 65 | 44.52% | | | |
| Other | 3 | 2.05% | | | |
| Vocational training | 11 | 7.53% | | | |
| Total responses | 146 | 100% | | | |

## 3. Results

### 3.1. General Understanding of What a Lawn Is and the Purpose of Lawns

There were a variety of different lawn definitions. For example, participants described lawns such as "A flat green groundcover popular in urban areas. Various types of turf"; "An area of maintained grass around a house or park"; "Urban green, well main-

tained turf represents wealth"; "Grass, turf, green non-native soft ground cover often used on verges. Takes a lot of water and mowing and edging"; "A green expanse of turf"; "A specific irrigated green space installed for fit for purpose uses such as play or wellness areas in both residential and non- residential areas"; "I consider lawn to be a space of grass used for ornamental purposes or various activities including, sports, social interactions and personal recreation"; "Tended turf grass that is mown and provides amenity for humans and pets and can be found in public spaces and homes".

Through coding, we generated seven categories identified based on a content analysis of lawn definitions: flat area; ground cover by grass; maintained; non-native vegetation; open space; recreational space; and turf grass (Figure 5). The most common definition related to a lawn was "a ground cover by grass".

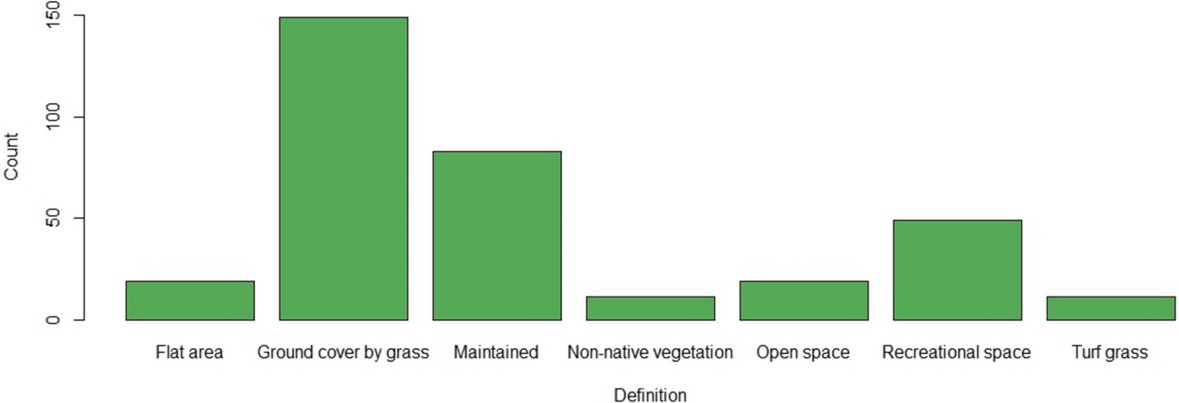

**Figure 5.** Categories identified based on content analysis of lawn definitions.

The other common definitions included concepts associated with a lawn being a surface that is "maintained" and used as a "recreational space" which demonstrates an understanding of the ecological and social essence (maintained grassy surface for recreation) of lawns by Perth dwellers. The common appearance of the words "non-native vegetation" in the definition also indicates an awareness about the origin of urban lawns in Australia (an introduced element from Europe).

The respondents most commonly identified the purpose of lawns as being for "recreational use", "aesthetics", "cooling effect", "ecosystem support" and "social status" (Figure 6).

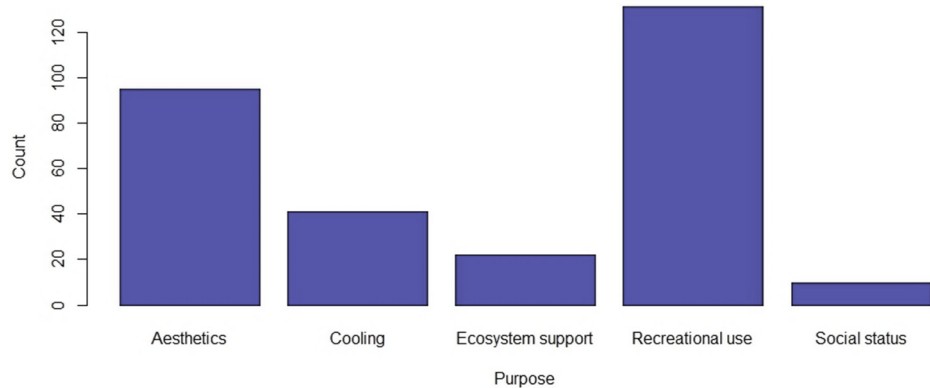

**Figure 6.** Purpose of lawns categories.

### 3.2. Uses of Lawn (Human and Non-Human)

3.2.1. Respondent's Use of Private Lawns

Around 49% of respondents often used their private lawns for passive recreation (sitting, socialising) followed by playing games (32%) and light exercise (26%), while only

7% of respondents used private lawns for vigorous exercise. More than 35% of respondents reported sometimes using their private lawns for light exercise, vigorous exercise, playing games and passive recreation (Table S2 and Figure 7).

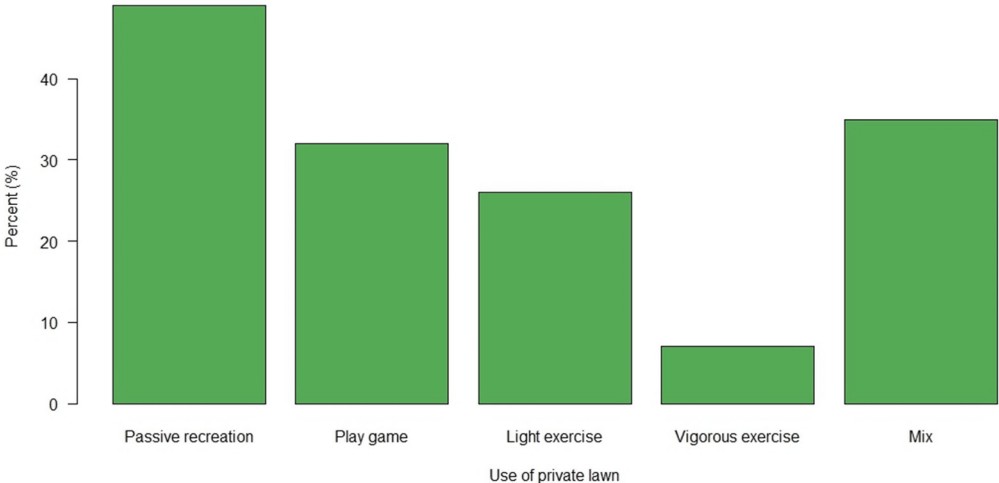

**Figure 7.** 'Often'-reported uses of private lawns for various activities.

### 3.2.2. Use of Public Lawns

Around 52% of respondents often used public parks for light exercise, 49% for walking pets, 41% for a transit area, 24% for passive recreation, 20% for vigorous exercise, 18% for social gatherings and 14% for sports (Table S3 and Figure 8).

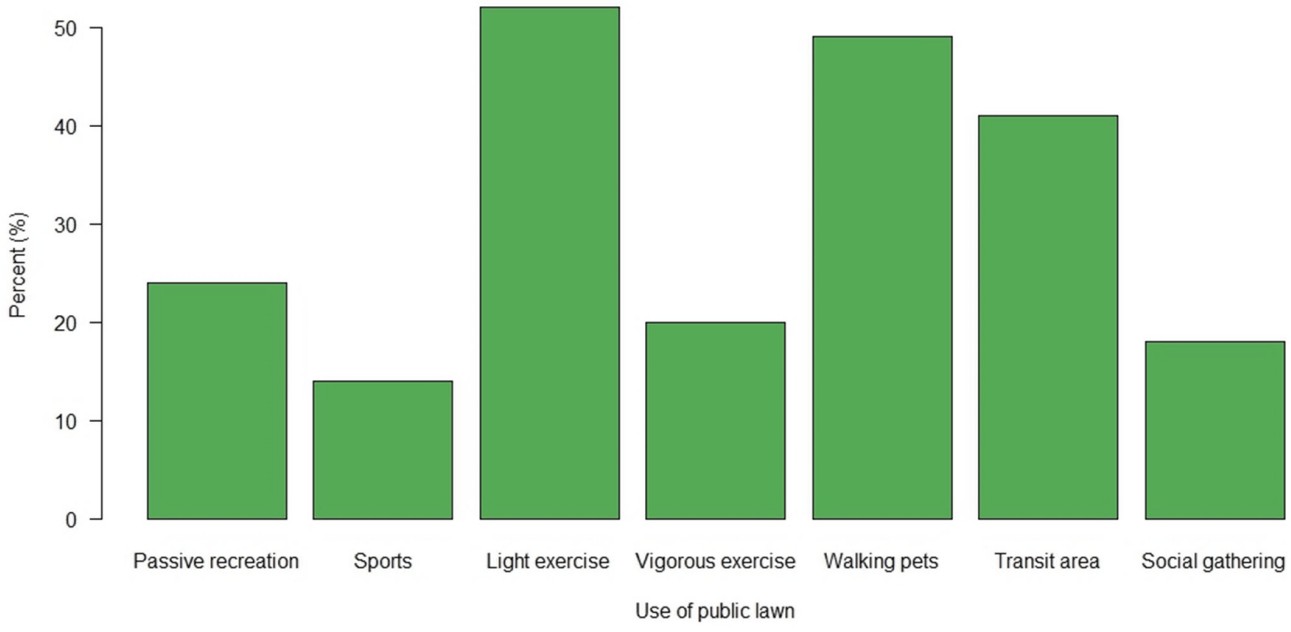

**Figure 8.** 'Often'-reported uses of public parks for various activities.

### 3.3. Lawn Use (Habitat) for Animals

Figure 9a represents respondent perceptions about whether the lawn is a good habitat for animals. About 37% of respondents reported that lawns are good habitats for animals, whilst more than half of the respondents reported that lawns are not good habitats for animals. Figure 9b shows that 84% of respondents reported that public and private lawns were also used by animals.

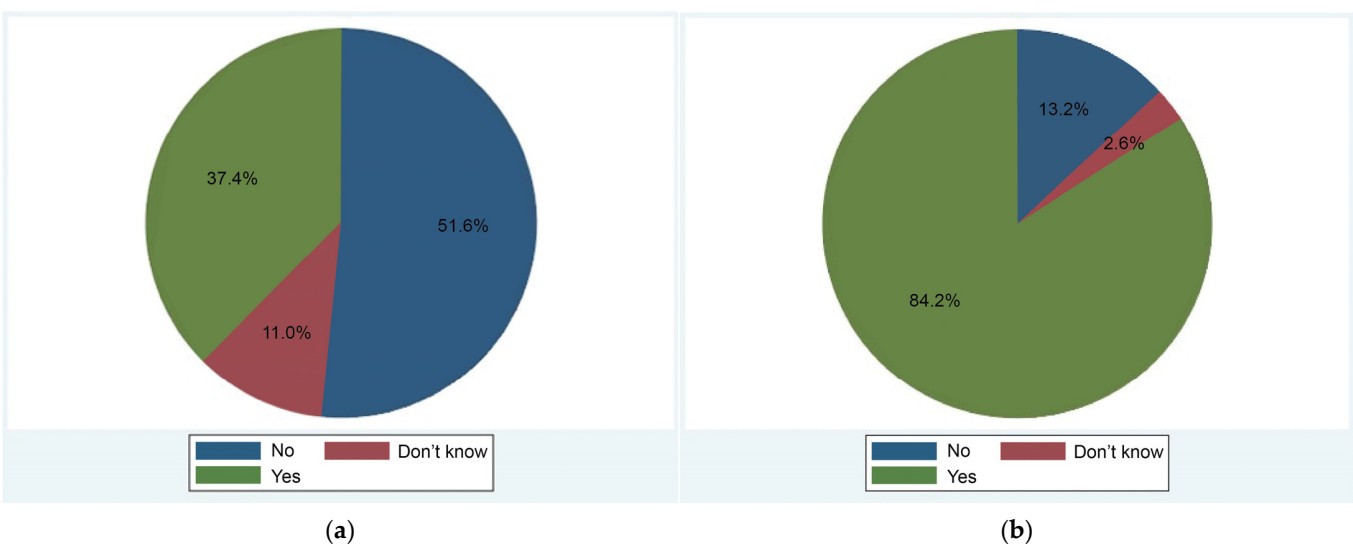

(**a**) | (**b**)

**Figure 9.** Perception about the lawn as a good habitat for animals (**a**) and animals using public or private lawns (**b**).

Respondents most often mentioned dogs, birds (ibis, magpie, willy wagtail, pink galah, corella, and Australian wood ducks) and sometimes kangaroos foraging on lawns (Figure 10). One of the respondents noticed "Native bees, particularly where clover is present".

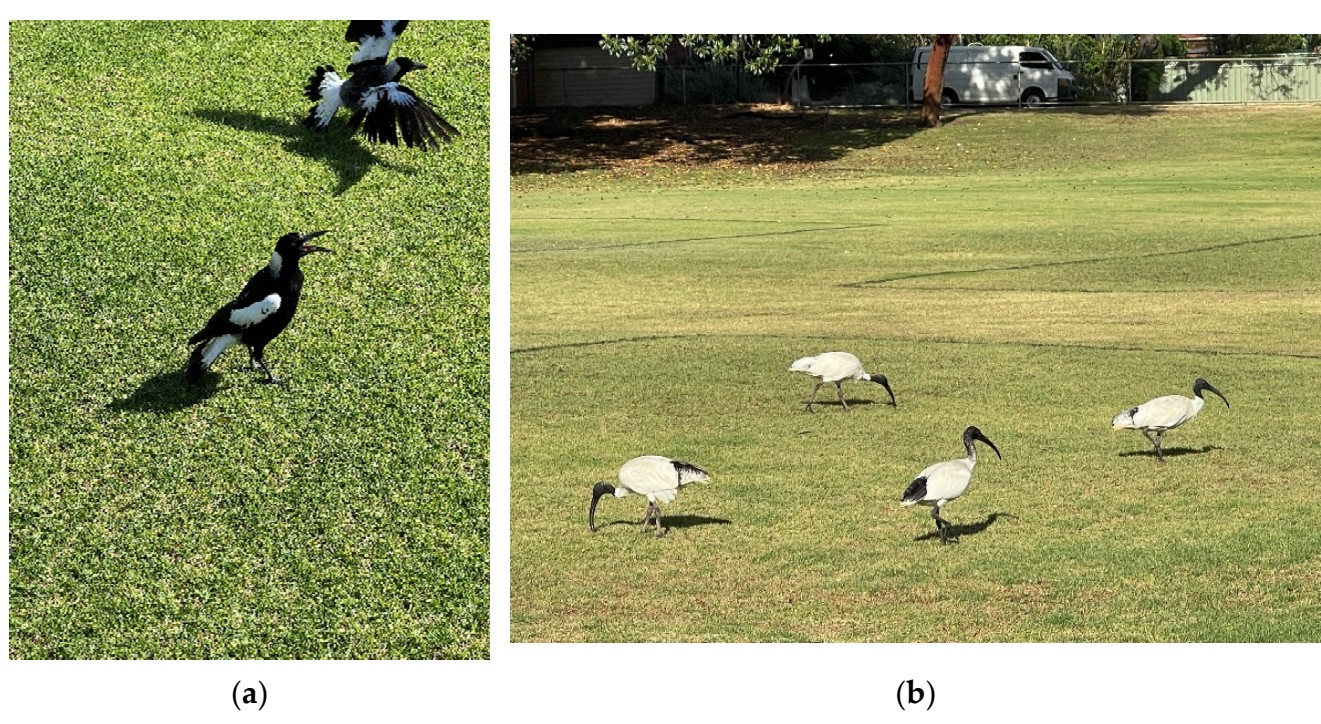

(**a**) | (**b**)

**Figure 10.** Magpies (**a**) and ibises (**b**) use public park lawns to forage in Perth.

*3.4. Maintenance and Perceived Quality of Local Lawns*

Private Lawn Maintenance

Around 39% of respondents mowed their private lawns once per fortnight followed by 37% of respondents who mowed their lawns once every month, whilst 3% of respondents had never mowed their private lawns since their establishment (Figure 11).

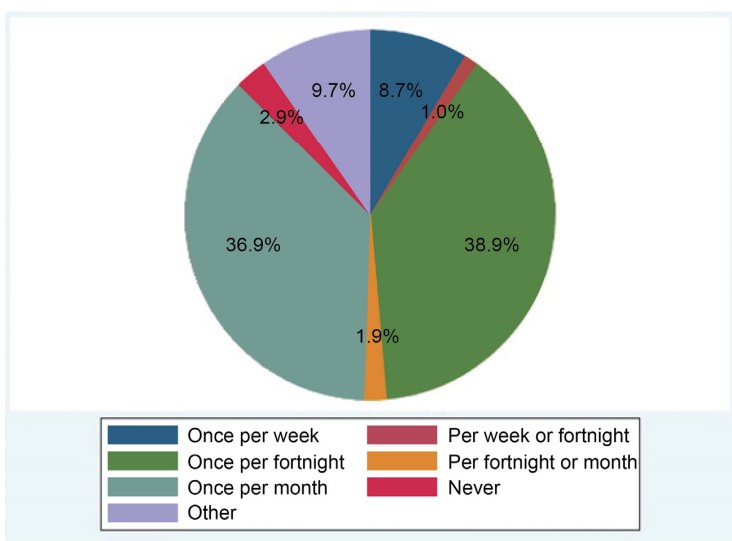

**Figure 11.** Frequency of mowing private lawns.

Around 45% of respondents preferred 3 to 5 cm tall lawns, and 40% of respondents preferred short lawns (less than 3 cm tall) (Figure 12).

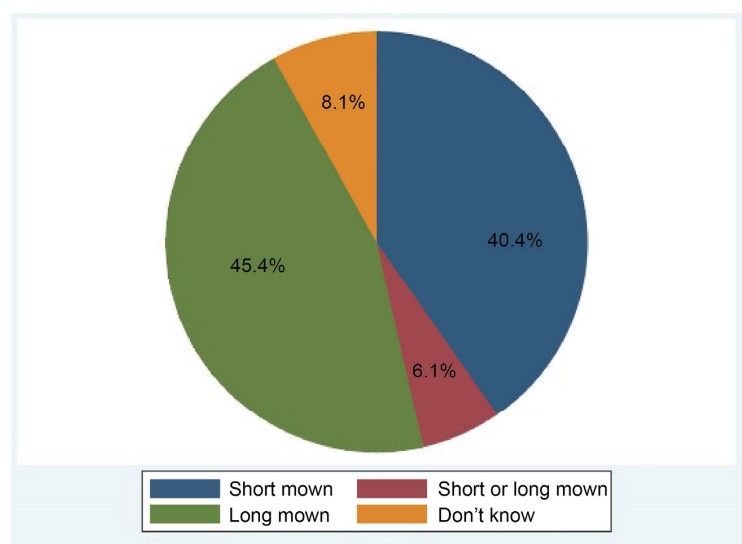

**Figure 12.** Height of private lawn.

Table 3 shows various lawn management techniques used by the respondents, and we found that 92 respondents used mowing methods followed by fertilisers (73 respondents), removed mown grass clippings (69 respondents) and added soil wetting agents (69 respondents as well).

**Table 3.** Summary of various lawn management techniques.

| Lawn Management Techniques | Obs. |
| --- | --- |
| Removing mown grass clippings | 69 |
| Fertilising | 73 |
| Applying herbicides | 32 |
| Adding soil wetting agent | 69 |
| Mowing | 92 |

Around 93% of respondents regularly watered their private lawns (Figure 13).

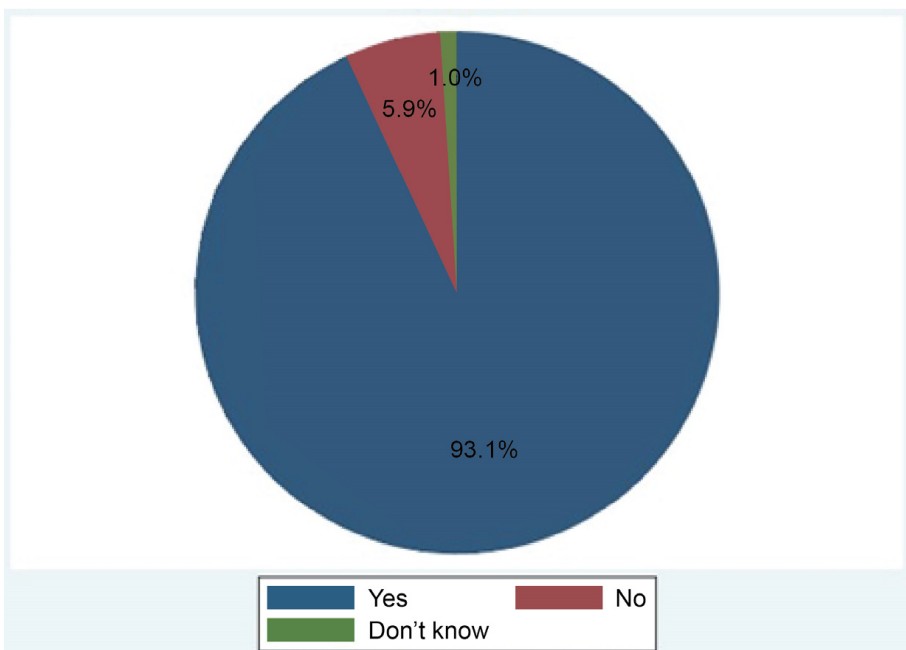

**Figure 13.** Regular watering of private lawns.

Only three respondents did not regularly water their private lawns because they thought lawns survive without watering (Table 4). These people mentioned the reasons why they do not irrigate lawns. For some, it was okay if the lawn died during dry summer months, while others believed that watering the lawn is a waste of water and that lawns survive without watering.

**Table 4.** Reasons for not watering private lawn.

| Reason for Not Watering Lawn | Obs. |
| --- | --- |
| No reticulation system | 0 |
| No time to hand water | 0 |
| Lawn survives without watering | 3 |
| Watering the lawn is a waste of water | 1 |
| OK if the lawn dies during dry months | 2 |
| Other | 3 |
| No sprinkler | 0 |
| I don't know how to keep my lawn healthy and green during summer | 1 |

*3.5. Perceived Quality of Lawns*

Around 37% of respondents were very satisfied with the quality of private and public lawns, which was followed by 33% of respondents who were somewhat satisfied and 16% of respondents neither satisfied nor dissatisfied. In addition, only 2% of respondents were very dissatisfied with the condition of private and public lawns (Figure 14).

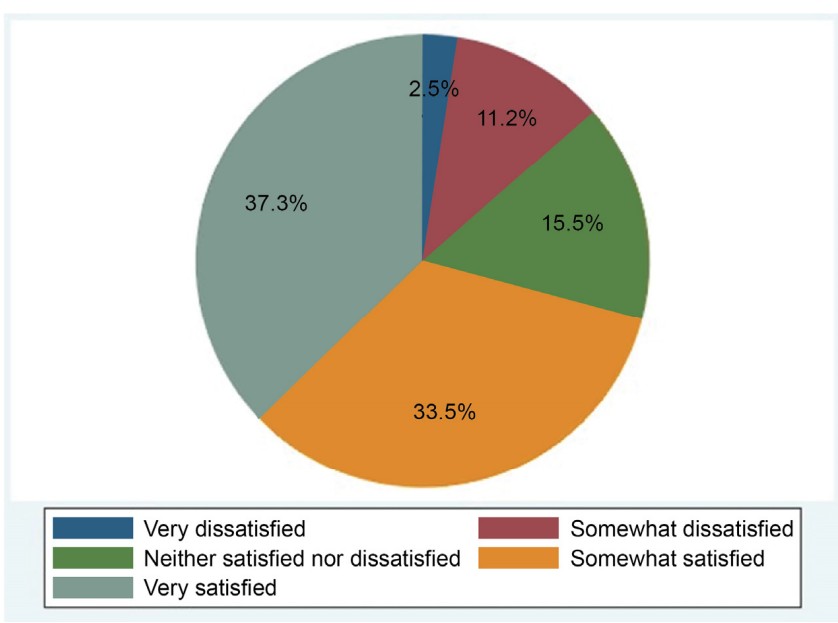

**Figure 14.** General impression about the quality of private and public lawns.

*3.6. Perceptions and Preferences Associated with Lawn Alternatives*

Most respondents indicated that Scaevola patches were good (61%), while lawns with turf grasses and Dichondra were the second most commonly positively rated image (57%). The least positively rated image was the old uncut high grass lawn (15%) (Table S4 and Figure 15).

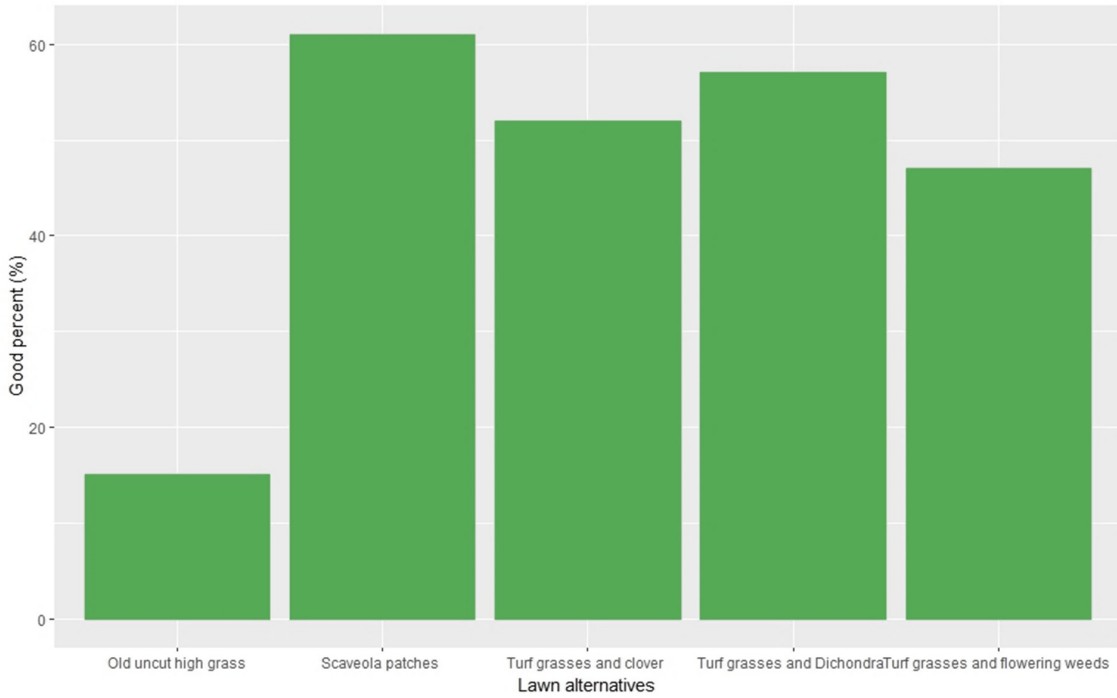

**Figure 15.** Positive rating of lawn alternatives.

Figure 16, Tables S5 and S6 present the ranking of five lawn alternatives that could be used in urban public parks. Around 35% of respondents ranked lawns with turf grasses and Dichondra above the other lawn alternatives. In contrast, old uncut high grass was ranked the lowest by 56% of respondents.

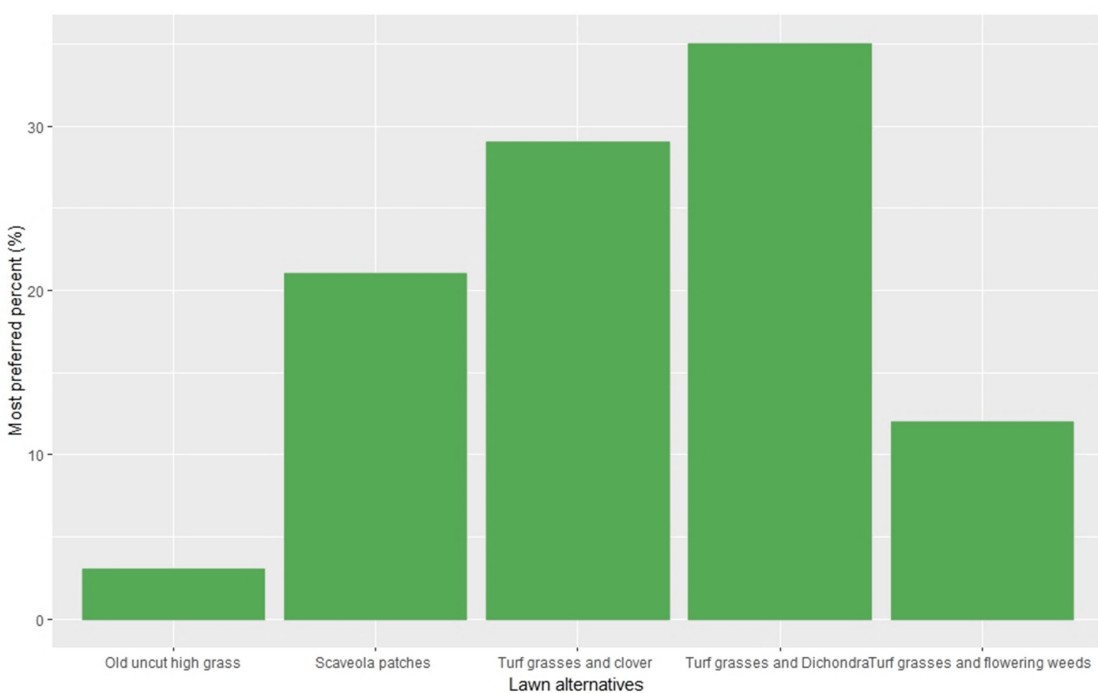

**Figure 16.** Comparative ranking of five lawn alternatives that could be used in public parks.

### 3.7. Summary of Logit Model Results

Our results show that the majority of respondents have private lawns. The age of the respondent, the type of house they live in, their education level, and retirement status had a significant effect on the ownership of private lawns (Table 5). Older respondents (aged 35–44, 45–54 and 55–64) had an increased log of odds for owning private lawns, respectively, compared to younger age groups (18–24). The age groups 25–34 and over 65 years had no significant effect on the ownership of private lawns (Figure 17).

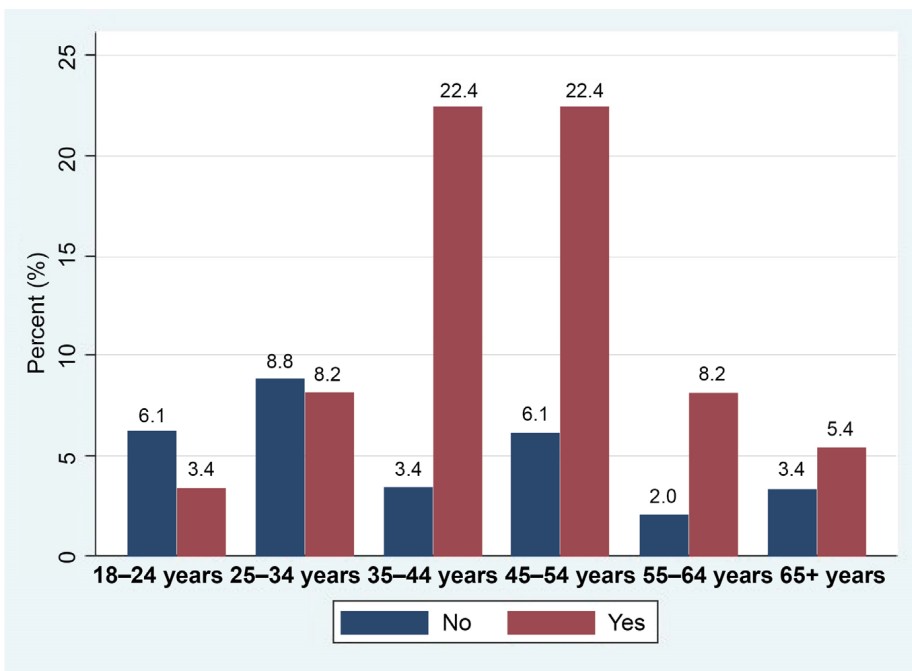

**Figure 17.** Ownership of private lawn by age group.

**Table 5.** Effect of socioeconomic variables on use of private lawn using a logit model.

| Variables | Coefficients |
| --- | --- |
| 25–34 years old | 0.474 |
| | (0.855) |
| 35–44 years old | 2.202 ** |
| | (1.078) |
| 45–54 years old | 1.863 * |
| | (0.981) |
| 55–64 years old | 2.612 ** |
| | (1.057) |
| 65+ years old | 1.003 |
| | (1.252) |
| Female | −0.724 |
| | (0.552) |
| Prefer not to say | −3.325 *** |
| | (1.276) |
| Flat/apartment/townhouse | −2.521 *** |
| | (0.658) |
| Other (please specify) | −0.570 |
| | (1.398) |
| Outside Australia | 0.643 |
| | (1.312) |
| Working part-time/casual | −0.211 |
| | (0.728) |
| Home duties | −0.512 |
| | (1.079) |
| Retired | −2.188 * |
| | (1.238) |
| Student | −0.623 |
| | (0.865) |
| University graduate | −1.291 |
| | (0.967) |
| University higher degree (postgraduate) | −1.215 |
| | (0.949) |
| Other | −4.388 ** |
| | (1.835) |
| Vocational training | −0.499 |
| | (1.335) |
| Constant | 1.893 |
| | (1.242) |
| Observations | 141 |

Robust standard errors in parentheses; *** $p < 0.01$, ** $p < 0.05$, * $p < 0.1$.

### 3.8. Preferences for Lawn Alternatives by Age Group

Figure 18a shows that at least 10% of the respondents from the age groups 25–34 years, 35–44 years and 45–54 years were more likely to prefer lawns with turf grasses and clover in our study. In addition, at least 9% of the respondents from the age groups 25–34 years, 35–44 years and 45–54 years preferred lawns with turf grasses and Dichondra (Figure 18b).

Around 16% of the respondents from the age group of 45–54 years preferred a biodiverse lawn with turf grasses and flowering weeds, followed by 9% for the age group of 25–34 years and the least (3%) for the older age group (Figure 19a). But for Scaevola flowering patches in dedicated spaces, around 17% of respondents from the age group of 45–54 years considered that it is a good idea, followed by 16% of respondents from the age group of 35–44 years, whilst younger respondents appreciated Scaevola patches the least (Figure 19b). The old uncut lawn option was unpopular among all age groups (Figure 20).

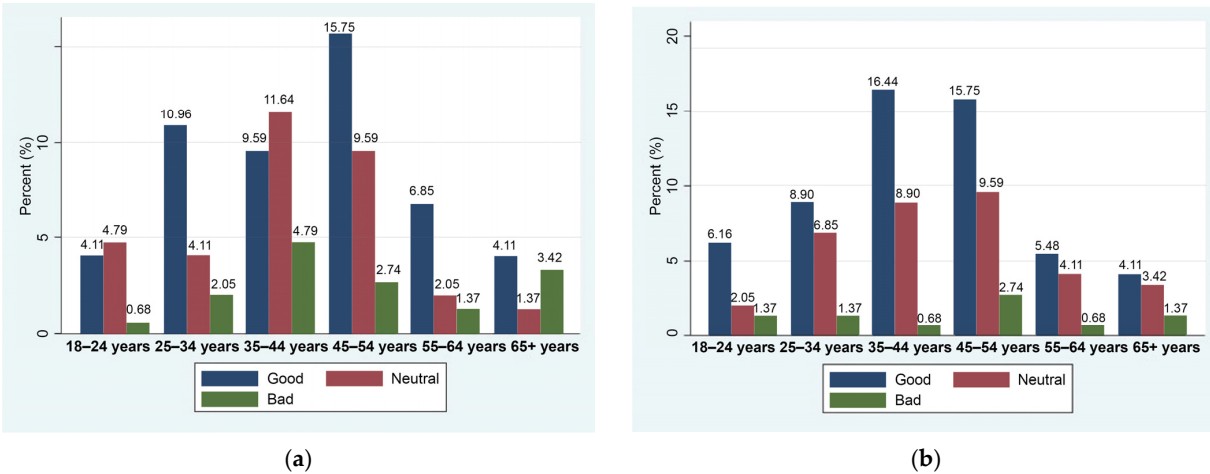

**Figure 18.** Preference for lawn alternative with turf grasses and clover (**a**) and preference for lawn alternative with turf grasses and Dichondra (**b**).

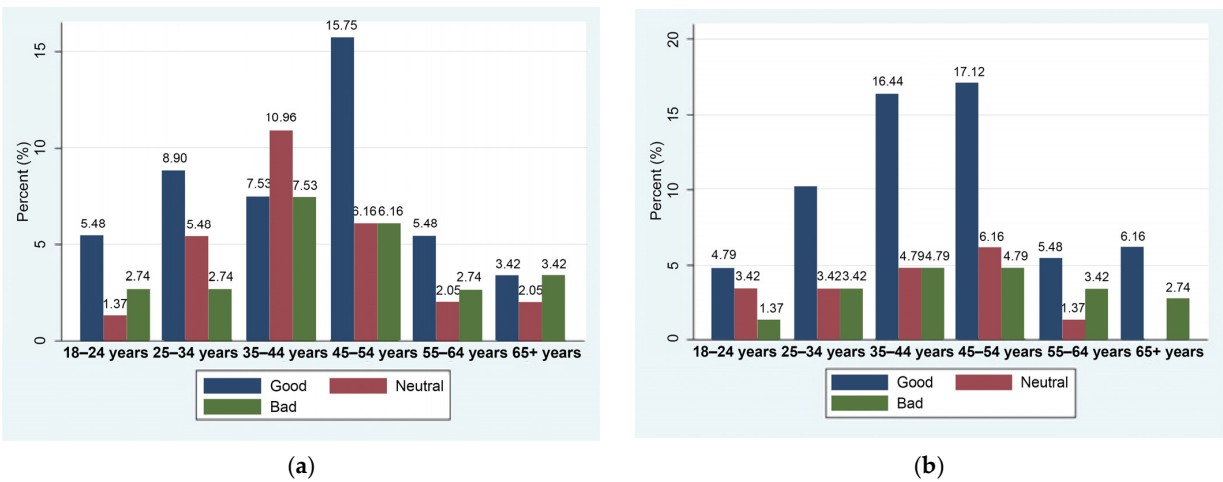

**Figure 19.** Biodiverse lawn with turf grasses and flowering weeds (**a**) and Scaevola patches in dedicated spaces (**b**).

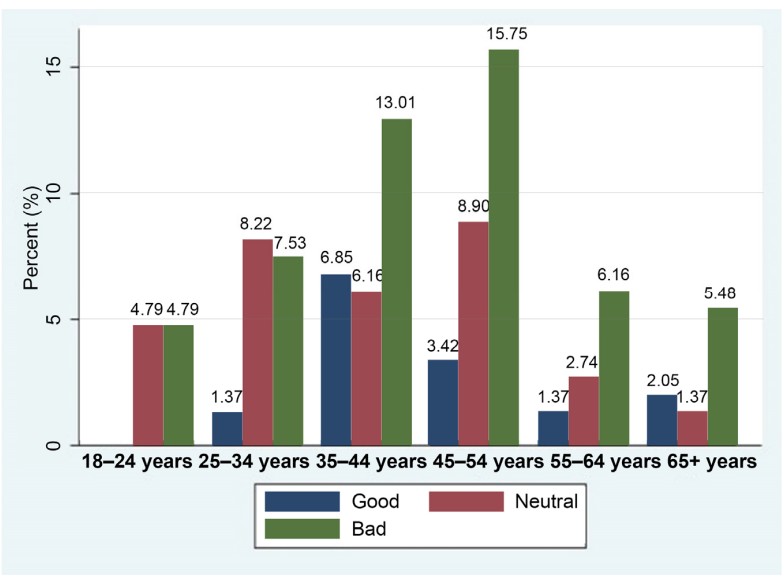

**Figure 20.** Old uncut high grass lawn.

### 3.9. Preferences for Lawn Alternatives by Lawn Definition and Purpose

The results showed no significant effect of lawn definition or purpose on lawn preferences (bad, neutral and good) of lawn alternatives (Figure 21).

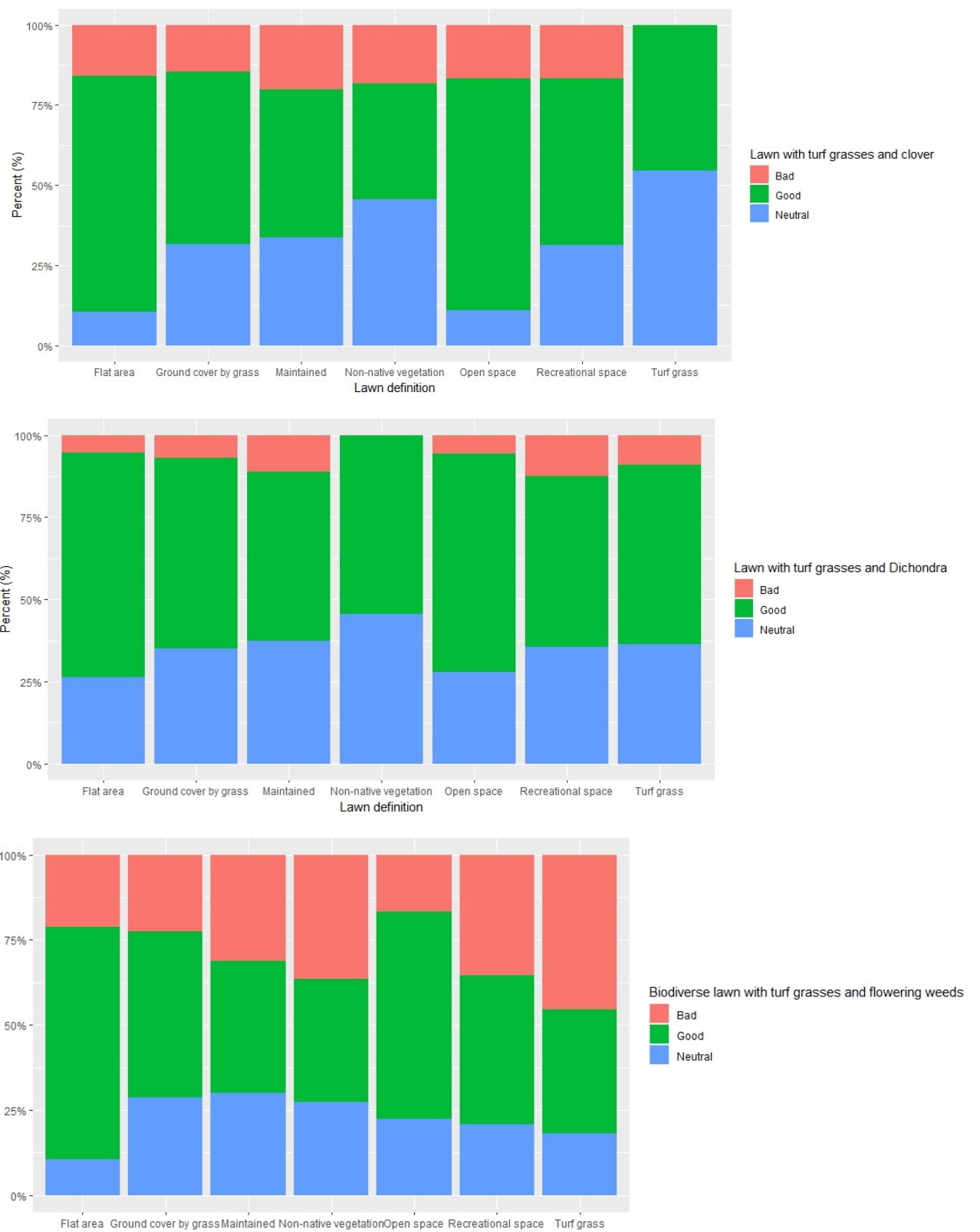

**Figure 21.** *Cont.*

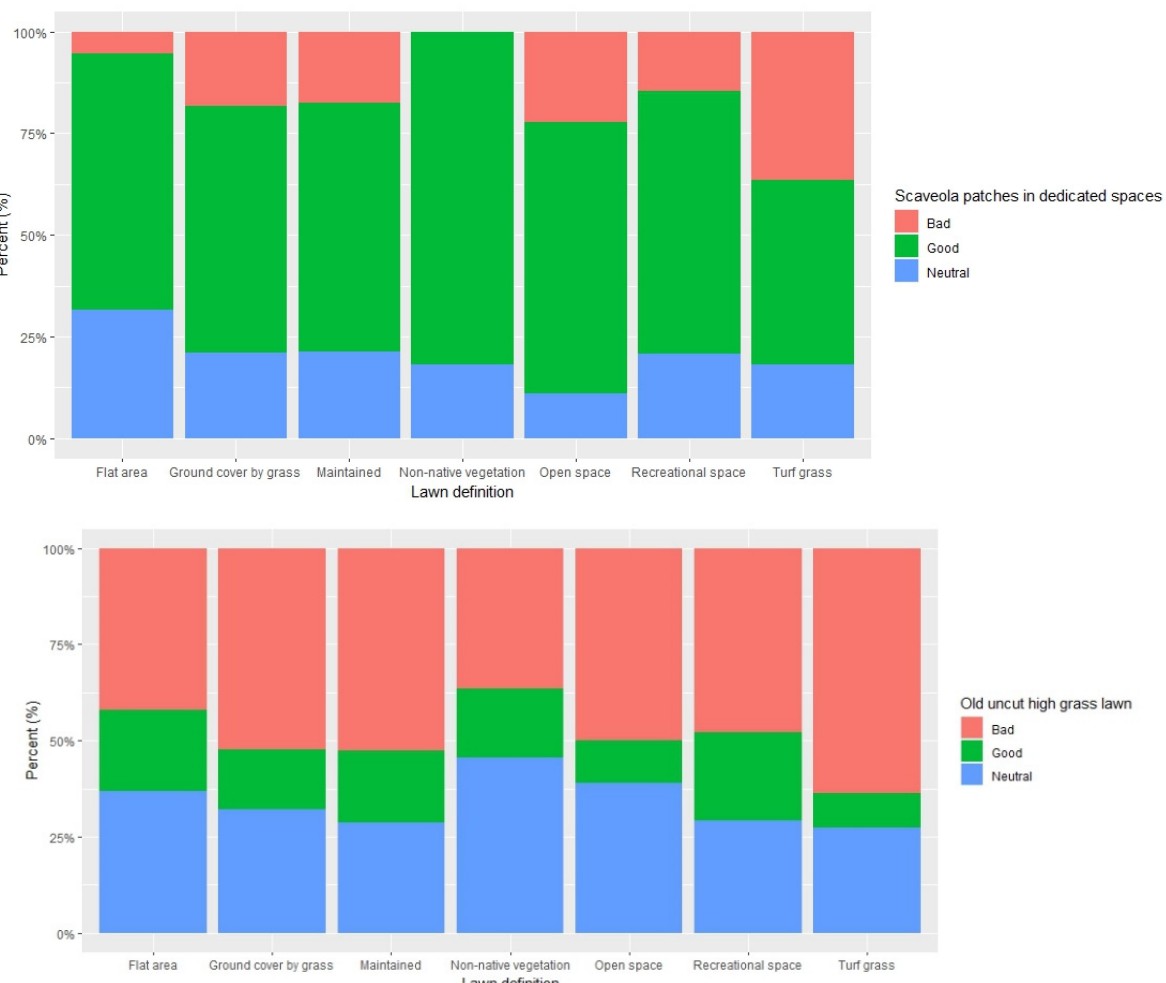

**Figure 21.** Definition of lawn groups combined with the alternative preferences for lawn alternatives.

Respondents who indicated the primary purpose of lawns being for social status (classical lawn) also positively rated lawns with turf grasses and clover. Turf grasses and clover were also highly rated among respondents, as it was the closest to the conventional lawn alternative option "turf grasses with Dichondra" (Figure 22).

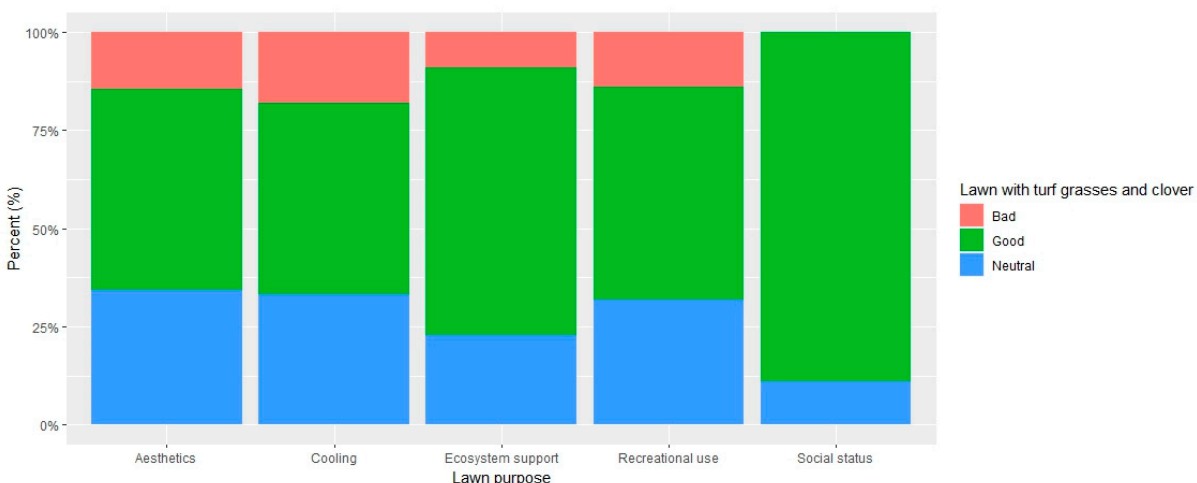

**Figure 22.** *Cont*.

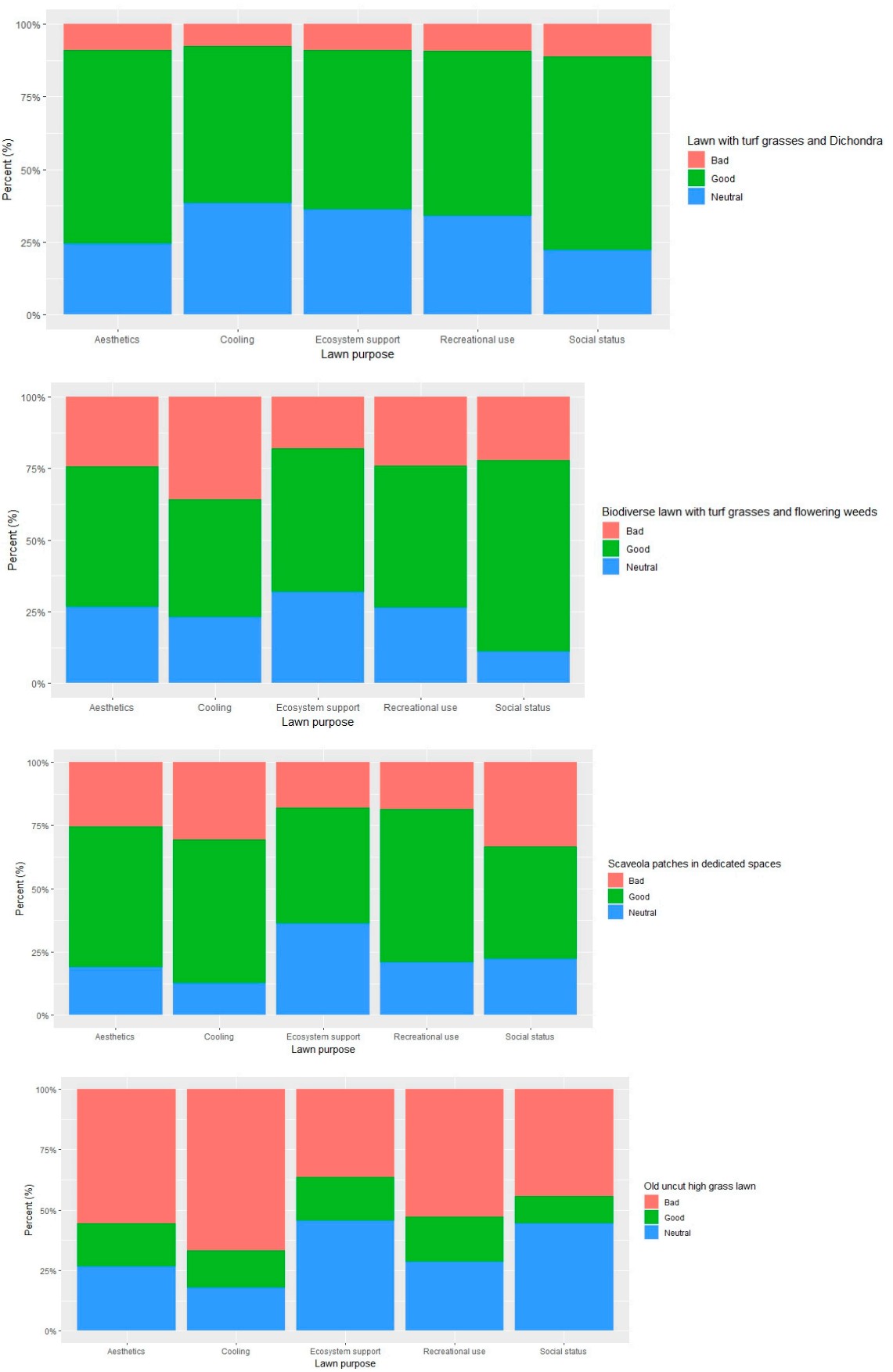

**Figure 22.** Purpose of lawns (overall) combined with the alternative preferences.

## 4. Discussion

### 4.1. Understanding of What a Lawn Is and Its Main Purpose and Use

The analysis of lawn definitions demonstrated that Perth's residents understand the ecological essence of a lawn, which is a ground covered by grasses that requires constant maintenance (irrigation, mowing, fertilising and weeding). Irrigation is one of the most important maintenance operations for Perth lawns because of long, dry and hot summers. Significant maintenance is required to create the desired grass-dominated plant community all year round. The use of terms such "non-native vegetation" and "open" spaces in the definitions also demonstrates the familiarity of some respondents with the environmental peculiarity of Australian urban spaces where all lawn grasses have been introduced from other parts of the world. The "public open spaces" (POS) is widely used in Australian governmental documents such as city masterplans and strategies [12], and knowledge of this term was indicated by many participants.

Some respondents provided quite detailed answers where they demonstrated knowledge of horticulture, landscape design and environmental sciences (e.g., "intensively managed area of turf grass and in between of hardscape and softscape"; "a variety of monocot cultured for its consistent pleasing green appearance" and "invasive species of grass"). This can be explained by the level of education of the respondents which is notably higher than the general population for our survey sample (81% university graduates and postgraduates). Other than the response to this question, the level of education generally did not have a significant effect on responses.

Another important term commonly used to define lawns was "turf grass". This corresponds with the overall tendency in Australia to use the term "turf" instead of "lawns". The term "turf" but not "lawn" dominates in all governmental documents (masterplans and urban development strategies) and landscape architecture and horticulture publications. The turf industry grows turf that is installed in public and private spaces. In Australia, the terms "turf" and "lawn" are synonymous. Even from an ecological point of view, turf is "the surface layer of soil with its matted, dense vegetation, usually grasses grown for ornamental or recreational use" [44]. Once turf (sod) is installed on the ground, it becomes a lawn—a particular grass-dominated plant community which is regularly maintained and which is "the issue of natural processes such as climate, growing of other species from existing seed banks, competition between species, developing of the microbial communities" [27] (p. 419). In the world's scientific and popular literature related to urban grass-dominated surfaces, "lawn" is the most commonly used term. In Australia, it was not until 1930 when turf farmers, who grow grass for dairy pastures, started to cut grass into strips and then sell them. Since that moment, the turf industry has developed, and an 'instant lawn' has become available for private homeowners and public green managers [45].

The association of lawns as a "recreation space" was among the most often mentioned categories. It also corresponded with the results of the use (purpose) of Perth lawns where "recreation" was the dominant category. From the very beginning, the lawn in Australia was designed to be a special amenity for sport and leisure.

The analysis of "purpose of lawn" answers demonstrated that respondents identified categories similar to the ecosystem services typology indicated in recent lawn research [3,5]. Recreational activity was most common, followed by the aesthetic qualities of lawns (both belong to cultural service), then cooling effect (regulating service), ecosystem support and social status. The appearance of "Ecosystem support" in participants' answers also indicated the respondents' awareness of environmental issues in urban environments such as urban heat and increasing dryness. The dominance of recreational use can be also explained by the climatic reasons and lifestyle of the people of Perth who are using the advantage of sunny and warm climates for recreational activities and can use lawns throughout the year. The positive effect of lawns for cooling urban spaces during the summer months in the Mediterranean climate has been indicated also for Santiago de Chile [46].

The high appreciation of the aesthetic value of lawns in Perth is connected to colonial heritage and nostalgia for the green colour of the British countryside. Urban "lawnscapes" were used in the battle against dust, sand and heat and as a tool for the "beautification" of the urban environment which should look more familiar and thus aesthetically pleasant [11]. The participants from Perth wrote such phrases pertaining to lawns as "aesthetic appeal", "aesthetic/beautification", "lawn is Australian Dream" and "turf is imbedded in Australian culture".

The beauty and tranquillity of the green colour provided by lawns are mentioned by urban citizens in other countries (e.g., Sweden and Germany), even in places where the green colour is quite dominant in all other types of urban green spaces and not only in lawns [5].

The use of the category "social status" that was identified by Perth's participants corresponded with the ecosystem service of "enhancing private property value, the symbol of prestige and power" that was identified for global lawns [1,5]. The aesthetic and social status are very connected in the case of Perth where green and well-kept private lawns indicated a local community status and an individual property status. The association of lawns with a "good citizen" and wealthy owners was reinforced during the modernist period in Australia when the minimalist well-kept low-cut lawns demonstrated the social norms and values of society [10]. Participants particularly noticed that lawns "provide eye-pleasing aesthetics to property" and that regarding having a lawn, "it represents wealth".

### 4.2. Use of Lawns for Various Activities (Human and Non-Human)

The analysis of activities in private and public lawns revealed that private lawns are used mostly for passive recreation, playing games and light exercise, while public lawns are mostly used for light exercise, walking with pets and as a transit area. Picnicking and playing sports also represented Perth's residents' favourite activities. Lawns in Perth cover a significant amount of space in public parks and school grounds. Interestingly, in a different part of the world, in the Swedish city of Malmö where public lawns are a prevalent type in public green spaces (parks and residential green spaces), typical activities have been shown to be very similar (walking, playing, jogging/exercising and transit) [5].

Lawn surfaces were created in place of native vegetation and bushlands, and the question of whether lawns are used by animals holds an important environmental and social message, given the increasing concern about losing native biodiversity in Perth. The participants provided quite detailed answers and mentioned native birds that often forage on lawn surfaces, including more generalist urban species as well as ibis and several parrot species (corella and galah) that have adapted to rely on grasses, clover and some weeds. People even noticed visits from kangaroos in some private lawns. Other creatures such as insects (including some native and honeybees) forage on flowering clover and weeds. Ants and lizards were noticed as well. Thus, these findings confirm that the lawn is seen as a specific novel biotope that attracts particular species that adapt to anthropogenic habitats. Dogs are also considered an important "user" of lawns in Perth.

Compared to European lawns, which have more native herbaceous plants (e.g., clover) that can bloom because of a more 'relaxed' mowing regime, Perth lawns are more homogeneous and less attractive for pollinators. However, Perth lawns are irrigated and attract more wildlife during the summer months compared to lawns in other parts of Australia where lawns are not irrigated during summer.

### 4.3. Maintenance and Perceived Quality of Local Lawns

The majority of respondents were satisfied with the quality of private and public lawns in Perth. For example, comments included that lawns are "kept green and tidy" and are "very well maintained and cared for by the LGA (Local Government Authority)". This "greenness" and good appearance is related to the regular irrigation of lawns in Perth in the summer months, frequent mowing (so that the grass is kept quite short to a length of less than 3 cm) the input of fertilisers and soil conditioners, which enable the retention

of moisture in the soil, and the provision of necessary nutrients to shortly cut grasses. Another important maintenance operation for Perth lawns is applying herbicides against spontaneously growing plants—weeds. The use of herbicides is a particularly controversial maintenance operation that is criticised by environmentalists as being harmful to people and the environment (water pollution). In many European countries, the use of herbicides in public green spaces is prohibited [5].

In our analysis, the demographics of participants that own private lawns showed that the groups of ages 35–44, 45–54 and 55–64 mostly own lawns and thus maintained lawns compared to the younger people. The older age groups own private lawns, potentially due to their greater financial stability, enabling them to be able to afford to buy a house with lawns, while the age groups containing younger participants may face financial constraints that make it challenging for them to own a house, including one with a lawn.

Among the respondents, there were also comments that the "Lawn is hard to keep in dry, hot Western Australia"; "The lawn closest to where I live is basically abandoned in summer and dries out"; "Public lawns can be overdone, e.g., some parks near me are excessively lawn-ey". People acknowledged the high level of care of public open spaces in Perth: "Having lived overseas I understand that well-maintained public spaces are a privilege. Upkeep of lawns is time-consuming and expensive, so we are very lucky to have the lawns are close to perfect". Recent hot summers in Western and Central Europe showed that places with no watering become dusty and fire-prone, thus turning ecosystem services into disservices [1,3]. However, this greenness is made possible at the expense of using water and a high input of resources. According to the Government of Western Australia Department of Water, 17% of urban water use goes to the irrigation of public parks. However, public park design and planning strategies have not planned to change the extent of lawns in sports ovals and some other grassed recreational spaces in the near future [17] because of the importance of lawns for human health during hot summers. The recent regulation on watering green spaces (including lawns) in Perth pushes for searching for more sustainable solutions, including for the maintenance of lawns, and more effective lawn surfaces that, on the one hand, can withstand heat and draught and, on the other hand, can create appropriate spaces for human recreation and even habitat provision.

*4.4. Perceptions and Preferences Associated with Lawn Alternatives*

In European countries, one of the most common alternatives to lawns is a meadow consisting of grasses and flowering perennials. The lawns can be turned into biodiverse meadows by decreasing the frequency of mowing which is often called "easy management" [6]. Recent research in European [6,47] and South American cities [46] confirmed that reducing the mowing frequency had a positive impact on plant and wildlife diversity. Because of the existence of sod, cut meadows could still be used for recreation similar to conventional lawns. There is also a series of alternative solutions in Europe and North America such as naturalistic plantings with a mixture of native and flowering exotic vegetation that aims to increase biodiversity and attract pollinators, particularly to improve the diversity of streetscapes, and decrease management input [9]. In Australia, there are only a few alternative solutions that have been tested, and all of them are related to the streetscape environment (verges). Such streetscapes can cover up to one third of public urban greenspaces in Australia [48]. Because of land ownership peculiarities and the limitation of resources, such places are not irrigated or mown, and this leads to the degradation of existing lawns. An alternative for streetscapes is planting low native plants instead of lawns. Recently, the Woody Meadow project inspired by English naturalistic plantings has become popular in Melbourne and Western Australia. These "meadows" use local native drought-tolerant species of shrubs and trees. A recent study on preferences for woody meadow plantings (using computer-generated images) compared to low-input non-irrigated conventional lawns in streetscapes showed preferences for woody meadows [48]. However, woody meadows cannot be walked on or used for recreation. Streetscapes and verges are the main targets in Australian cities for "rewilding" and native plant introduction (the substitution

of lawns) which have also been studied from a sociological point of view (social norms and ecological values) [49,50].

Our alternatives used in the survey suggested several options that can be used as possible solutions in the open public spaces of Perth. These scenarios used a low-growing native ground cover such as *Dichondra repens* that is already spontaneously growing in some lawns (imbedded into lawn grass *Cynodon dactylon* (couch)) and that can withstand some human traffic. Another option is a monoculture of low-growing (up to 20 cm) and flowering Scaevola cultivars that could be used in some areas of private gardens and dedicated public open spaces. Another suggestion is a more biodiverse version of lawns with commonly used grasses and clover (*Trifolium repens*) that are low-growing but can flower and thus provide a forage for pollinators. One of the options is also based on existing examples in Perth where lawn grasses are embedded with flowering annual weeds and, in spring, create low meadow-like lawns. Finally, an "uncut lawn" mimics the existing practice in some European municipalities to let lawn grass grow tall and be cut only once a season.

The findings indicate that respondents liked the Scaevola alternative and the lawn with Dichondra. The position of Scaevola with bright lilac flowers that resembles a beautiful carpet corresponded with the finding of public appreciation of naturalistic, grass-free and pictorial meadows in Europe because of an increase of attractive colourful flower displays in urban environments [51]. Scaevola flowering patches were most appreciated by the middle age groups which might be explained by the growing of environmental movements in Perth and the popularisation of rewilding private gardens by growing native plants. Surprisingly, the option of alternatives with flowering clover and the option of lawns with flowering annual weeds were considered as a possibility by all age groups. However, the older group of people had the least appreciation of non-traditional options for lawns. An indication of the necessity for more careful placement of lawns in open spaces as well as searching for more effective and smart maintenance was indicated in additional comments and definitions such as "Lawn creates a space that is multifunctional and reduces heating in an urban environment. However, lawns should not be used unnecessarily in spaces where the above-listed activities do not take place".

However, the comparative ranking of the five lawn alternatives revealed that participants clearly preferred those alternatives that resembled a standard lawn such as the lawn grass with Dichondra or clover. Furthermore, these types of lawn alternatives were far more preferred by respondents who defined the purpose of the lawn as being for social status. This indicated the continuing importance of lawns as an attribute of Perth suburban communities and as a very practical element in creating "culturally familiar environments" [11]. Taking into consideration that recreation is the most valuable use of lawns in Perth, the crucial criterion for alternative lawns would be a durable surface for walking.

Most respondents (of all age groups, education and gender) rated and ranked old uncut high grass negatively. This result reflected the history of Western Australian lawns. High grass vegetation resembles European pasturelands but is not presented in Western Australian native biomes. Instead, native shrubs and woodland vegetation with a complex "messy" structure and olive green and grey foliage contrast with the lawn. For lawns, people who use prefabricated "ready" turf that is regularly cut, and where all other plants (different kinds of weeds) are eliminated, uncut lawns are associated with a lack of care rather than an alternative biodiverse option. In addition, high grass in the Australian urban environment might be associated with the possible presence of dangerous creatures such as snakes and spiders. In Europe and the USA, the "cues to care" option where a strip of cut and manicured lawn borders "messy" meadow plantings are one of the preferable options in urban landscapes that can solve the dichotomy of introducing native nature into urban environments [5,52].

When given the opportunity to include additional comments in the survey, participants also expressed concern about using synthetic lawns, which is reflected in the latest urban phenomenon—using plastic and fake lawn nature [53,54]. Comments like "Plastic lawn is

an abomination" demonstrate the environmental and social awareness of Perth's citizens about the negative impact of synthetic turf on human health and the environment.

### 4.5. Limitations of This Research

This research had several limitations to consider. Firstly, while online survey methods enabled distribution to a dispersed sample population across the Perth metropolitan area, they are prone to self-selection bias, where respondents tend to be people who have ready access to the internet, are interested in the topic and are motivated to express their views. This is evident in the relatively higher education level of respondents compared to the census data for the general Perth population, even though the remaining respondent demographics were similar. While there was a bias in the sample towards university education, the analysis indicated that the education level was not significantly associated with lawn preferences, suggesting this bias did not influence the general findings of this research. Secondly, asking respondents to rate images on a screen to ascertain preferences is a common and accepted technique, but this has limitations in terms of preferences solely being determined by visual appearance. Designing a field experiment in which respondents directly experience different lawn alternatives in person may provide additional research insights. Online images are limited to purely visual cues, while a field experiment may add other components such as touch, sound and smell that could influence preferences. Designing a field trial or experiment would require significant resources to ensure a representative sample but would provide additional insights that could complement the online survey findings.

### 5. Conclusions

Despite the wide distribution of lawns in open public and private spaces in Perth, they have not been comprehensively investigated from a sociological point of view. Our study identified and analysed public views on the visual appearance, maintenance and uses of urban lawns and the visions on lawn alternatives in private gardens and public parks in Perth based on online surveys. The sample generally aligned with the greater Perth population profile, except in terms of the level of educational achievement. However, the education level did not have a significant effect on preferences for lawn alternatives.

Lawn is an important urban legacy in Western Australia with almost 200 years of colonial history. Participants value their lawns and recognise their importance in the urban landscape for a variety of recreation activities, cooling the environment and aesthetic benefits. At the same time, participants demonstrated an environmental awareness of lawns and the necessity of revisiting the existing planning and maintenance routine based on irrigation and intensive mowing by considering several alternative solutions. While valuing new solutions such as Scaevola patches in dedicated areas and "weedy lawns", participants still preferred alternatives closest in appearance to a conventional lawn (e.g., lawn grass with Dichondra and lawn grass with clover). There will be always a need in private and public green spaces for lawns that can provide durable surfaces for recreational activities such as sports, playing and picnicking.

There is an opportunity for further research into a "blended model" of urban lawns that would retain their essence as durable surfaces but could contain grasses and herbaceous and/or ground cover species that can withstand heat and recreational trampling and provide other ecosystem services such as a habitat provision for biodiversity. There also should be a search for more drought-tolerant species that might be considered more species-diverse versions of turfs. Another important direction can be the experimental planting of native Western Australian grasses and trying to create versions of native grass-dominated meadows. Several countries (e.g., Sweden, China, France) have initiated demonstration trials in botanic gardens and university campuses that test different options of alternative solutions and residents' perceptions of and preferences for alternative lawns [27,36].

The search for effective alternative solutions should ideally be based on a holistic view of how and where different lawn types should be located in urban green/open spaces and

what their design and management strategies and characteristics are. This direction would support already existing initiatives in Perth such as hydrozoning, ecozoning and verge rewilding. The results of our research on the preferences and expectations for lawns in Perth can be also used in developing planning and designing guidelines for urban public open spaces and also for the Western Australian turf industry, cities and municipalities.

**Supplementary Materials:** The following supporting information can be downloaded at: https://www.mdpi.com/article/10.3390/land13020191/s1, Table S1: List of questions used for the survey; Table S2: Use of private lawn for various activities; Table S3: Use of public park for various activities; Table S4: Lawn alternatives that could be used in public parks; Table S5: Ranking of lawn alternatives; Table S6: Ranking of lawn alternatives.

**Author Contributions:** Conceptualization, M.I. and M.H.; methodology, M.I., M.H., F.M. and A.K.C.; validation, M.I. and M.H.; investigation, M.I. and M.H.; resources, M.I. and M.H.; data curation and interpretation of results, M.I., M.H. F.M. and A.K.C.; writing—original draft preparation, M.I.; writing—review and editing, M.I., M.H., F.M. and A.K.C.; visualization, M.I., A.K.C. and F.M.; supervision, M.I.; project administration, M.I.; funding acquisition, M.I. All authors have read and agreed to the published version of the manuscript.

**Funding:** This research was a part of the University of Western Australia research project "Lawn as an ecological and cultural phenomenon in Perth" (2022/GR000551) funded by the following collaborators: the City of Rockingham, the City of South Perth, the Turf Growers Association of Western Australia, Turf Producers Australia, the Department of Water and Environmental Regulation, the Water Corporation, Murdoch University, Stratagreen, ArborCarbon, and Syrinx Environmental.

**Institutional Review Board Statement:** The Human Ethics approval for this research was obtained from Murdoch University Human Research Ethics Committee (2022/038) and the University of Western Australia Human Research Ethics Committee (Reference number 2022/ET000260).

**Data Availability Statement:** The data presented in this study are available on request from the corresponding author. The data are not publicly available due to privacy restrictions as governed by research ethics approval conditions.

**Acknowledgments:** We thank Daniel Martin (UWA) for preparing the data on the coverage of lawns in Perth's 'Urban Zone'. We appreciate the participants of the online survey for their valuable time and contribution. We also would like to thank the reviewers for their comments and suggestions.

**Conflicts of Interest:** The authors declare no conflicts of interest.

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
