# Peer review of "The Lawn as a Social and Cultural Phenomenon in Perth, Western Australia"

_land, doi:10.3390/land13020191_

Round 1

Reviewer 1 Report

Comments and Suggestions for Authors

A well prepared paper covering an  aspect of increasing importance. Overall it is fine but there are a few aspects that need to be worked on.

The abstract of the paper is far too long and contains too much detail. Please reduce it in length.

Use of the term "lawn" seems to be written without an article so grammatically there need to be some corrections.

Figure 1 shows examples of less-intensively managed lawns but it would be good to show some examples of classical mown lawns as a reference. More could also be made of the aesthetic aspect of lawns - the smooth, even texture and fresh green colour which sets off the other garden plants.

Given that the study is specific to Perth, it might be helpful to pick out any aspects of climate or other aspects which are unique to the city compared to the rest of Australia.

The objectives and aims of the study could be more clearly presented, especially a set of research questions.

It is better to present the characteristics of the respondents in the methods section rather than the results, because they are not strictly speaking results.

The results section is very long and full. Is it necessary to include both tabular and graphical presentation of the same data?

The discussion could be better focused if the research questions were more specific. The fact that the majority of the respondents had higher education qualifications and therefore could give an unrepresentative picture needs to be stressed. It is likely that such respondents are more environmentally aware and maybe have more knowledge which may affect the results.

The limitations of the research are not mentioned. There must be some, so please add them.

Comments on the Quality of English Language

The language is good apart from a few typos and the point about how to use the term "lawn" mentioned in the review text

Author Response

[LAND] Manuscript ID: land-2813308

Answers to the comments of Reviewer 1

We thank reviewer for looking at our manuscript and for the thorough feedback and very useful recommendations. Our responses to the reviewer's questions are below.

  1. A well-prepared paper covering an aspect of increasing importance. Overall it is fine but there are a few aspects that need to be worked on.

We thank the reviewer for the recognition of our efforts and positive comments.

The abstract of the paper is far too long and contains too much detail. Please reduce it in length.

We reduced the length of the abstract. Now it is 254 words.

Use of the term "lawn" seems to be written without an article so grammatically there need to be some corrections.

We checked the use of the article “the” within the text and fixed it.

Figure 1 shows examples of less-intensively managed lawns but it would be good to show some examples of classical mown lawns as a reference. More could also be made of the aesthetic aspect of lawns - the smooth, even texture and fresh green colour which sets off the other garden plants.

We included the photo of the conventional lawn showing the aesthetic function of the lawn in Figure 1. We also added the following sentences that highlighted the importance of the aesthetic aspect of lawns. Lines 74-75:

“One of the most important aesthetical features that make lawns attractive to urban citizens is the green colour and its tranquillity and beauty. The lawn is an important element of landscape design….”

At the end of the Introduction section (lines 146-153), we also included a new paragraph:

“The decrease of water use is the red thread of new studies on public preferences for different landscape design scenarios that could decrease the irrigation needs in public parks. One example is redesigning the park grounds and substituting some watered lawn areas with draught-tolerant native vegetation, groundcovers and mulch [17]. One particular type of lawn in Perth received more attention as a potential for alternative solutions. These are lawns located on verges (areas between streetscapes and private property boundaries) [18]

In Australia “true” alternative solutions to lawns (surfaces that are like lawns, e.g., grass-dominated communities that can accommodate tramping and regular mowing) could be inspired by different local grassy ecosystems and even by “hybrid” models where native herbaceous species are blended with lawn’s grasses. “

Given that the study is specific to Perth, it might be helpful to pick out any aspects of climate or other aspects which are unique to the city compared to the rest of Australia.

Thank you for this comment.  We included an additional paragraph in the method’s “Case study” section that compared Perth with other Australian cities (Lines 193-203) and also included a paragraph about unique Perth’s hydrology.

“Perth's climate is different from the climates of major cities on the east coast of Australia. For example, Sydney is located in a humid subtropical climate with mild and cool winters, warm and hot summers, and no dry seasons. With an average rainfall of around 1175 mm a year. Melbourne is located in a temperate oceanic climate with an average rainfall of around 650 mm a year and even a rain distribution pattern during the year.  The closest climate among Australian cities is Adelaide in South Australia which is also located in a Mediterranean type of climate with the average rainfall around 550 mm a year. However, Perth is hotter than Adelaide in the summer and Adelaide is cooler in winter.

Below Perth, there are three layers of aquafers: superficial, the deeper Leederville Aquifer, and lower, the Yarragadee Aquifer with ancient waters of 40 000 years old with an extraordinary capacity of water (1,000 cubic kilometers of water). No other Australian city and many cities around the globe have such extensive aquafers. This unique hydrological system supports Perth’s rivers, wetlands, waterways, and diverse native vegetation. It supplies an important part of potable scheme water [23].  For many decades Perth’s green spaces including lawns had a privilege to be watered several times a week.”

Line 207-209: “The mixture of groundwater (47%) and desalination water (53%) makes Perth unique compared to other state capitals in Australia that are highly dependent upon surface water [23].”

The objectives and aims of the study could be more clearly presented, especially a set of research questions.

Thank you for this valuable comment. We included the research questions and also provided a stronger argumentation about the necessity and importance of this research at the end of the Introduction (Lines: 166-175: “Compared to Europe, lawn in Australia never was an object of a separate scientific study neither as a specific urban biotope or social phenomenon nor as an aesthetical element of public and private green spaces. This study aims to fill the gap in recognising lawns as a complex socio-cultural-ecological entity in Australia. The main research question is “How do people of Perth, Western Australia define and use private and public lawns as well as understand their role in an urban environment?” Another research question is “What are the current social values and preferences of different socio‐economic groups of people toward existing traditional lawns and lawn alternatives that introduce new species, designs and management strategies?” One sub-question of this study is “What is the current maintenance and perceived quality of local lawns in Perth?.” This question aims to reveal the existing maintenance routine of urban lawns and in perspective suggest more sustainable and economical approaches. This particular question was included due to the recent decision to change the sprinkler roster for scheme water users in Perth and Mandurah, reducing from three days per week [20]”.

It is better to present the characteristics of the respondents in the methods section rather than the results, because they are not strictly speaking results.

Thank you for the valuable advice. We moved the characteristics of the respondents to the end of the Method section. Lines 308-318.

The results section is very long and full. Is it necessary to include both tabular and graphical presentation of the same data?

Thank you for your valuable advice. We removed tables from the main text and included them in the Supplementary material.

The discussion could be better focused if the research questions were more specific. The fact that the majority of the respondents had higher education qualifications and therefore could give an unrepresentative picture needs to be stressed. It is likely that such respondents are more environmentally aware and maybe have more knowledge which may affect the results.

We included the research questions at the end of the Introduction and in the Discussion section we discussed each question and theme that was identified during the research (e.g. Understanding of what lawn is and its main purpose and use; Use of lawns for various activities (human and non-human); Perceptions and preferences associated with lawn alternatives.). We discussed the issue that “the majority of the respondents had higher education qualifications” in the Limitation section.

The limitations of the research are not mentioned. There must be some, so please add them.

We included the Limitation section (Lines 692-704)

This research had several limitations to consider.  Firstly, while online survey methods enabled distribution to a dispersed sample population across the Perth metropolitan area, they are prone to self-selection bias, where respondents tend to be people who have ready access to the internet, are interested in the topic and are motivated to express their views. This is evident in the relative higher education level of respondents compared to the census data for the general Perth population, even though the remaining respondent demographics were similar.  While there was a bias in the sample toward university education, the analysis indicated that education level was not significantly associated with lawn preferences, suggesting this bias did not influence the general findings of this research.  Secondly, asking respondents to rate images on a screen to ascertain preferences is a common and accepted technique, but has limitations in terms of preferences solely being determined by visual appearance. Designing a field experiment in which respondents directly experience different lawn alternatives in person may provide additional research insights.  Online images are limited to purely visual cues while a field experiment may add other components such as touch, sound and smell that could influence preferences. Designing a field trial or experiment would require significant resources to ensure a representative sample but would provide additional insights that could complement the online survey findings.

Comments on the Quality of English Language

The language is good apart from a few typos and the point about how to use the term "lawn" mentioned in the review text.

We checked the revised manuscript and fixed the errors.

Reviewer 2 Report

Comments and Suggestions for Authors

This manuscript collects the public's understanding and opinions of lawns in the form of an online questionnaire, and organizes the collected answers to provide an in-depth analysis of lawns in Perth, Western Australia. The overall content is very informative, but there are still more questions that need to be addressed.

1The content of the manuscript is redundant and not condensed enough. In the part of the introduction of the study area, a lot of space is spent on the introduction, and I personally feel that some contents not related to this study can be deleted appropriately.

2The overall point is unclear. After reading this manuscript I don't understand what the purpose of this article was. You show too many graphs in the results section, which reflects the amount of work you did. But at the same time it doesn't express what the significance of this data reflects.

3The methodology section of the manuscript is so small that a data analysis module is totally insufficient.Moreover, the source of data is too subjective in the form of online questionnaires only, and some other supporting data need to be added to enhance the persuasiveness of the argument.

4The presentation of figure 5 is a bit poor, and the distribution icons can be modified, and the corresponding legend, compass, scale and other elements can be added.

Comments on the Quality of English Language

The clarity and flow of language in the manuscript is good, with clear and concise sentence structure and few spelling errors. However, there are some areas where terminology use could be made clearer to make sentences more direct. Overall, the language style and document structure are as expected, but some revisions in grammar and terminology are still recommended to improve the quality.

Author Response

We thank .the reviewer for looking at our manuscript. Here are our responses to the reviewer's questions.

1The content of the manuscript is redundant and not condensed enough. In the part of the introduction of the study area, a lot of space is spent on the introduction, and I personally feel that some contents not related to this study can be deleted appropriately.

We are unsure what revisions the reviewer is specifically suggesting with this comment. The comment seems to be more of a general subjective opinion about the quality and content of the manuscript.  We are keen to address the reviewer’s concern in order to improve the manuscript. Providing more concise information, such as the line numbers associated with the sections requiring revision, and what specifically needs to be changed would be helpful. 

2The overall point is unclear. After reading this manuscript I don't understand what the purpose of this article was. You show too many graphs in the results section, which reflects the amount of work you did. But at the same time, it doesn't express what the significance of this data reflects.

The aim of the paper is stated in the Introduction, Discussion and Conclusion sections, the significance of the data is indicated in the results, discussion, and conclusion sections. Some more specific suggestions on how we can improve the paper would enable us to best address the reviewer’s concerns.

We have revised and improved the manuscript to be sure that the main aim and the results are clearly presented and discussed.

3The methodology section of the manuscript is so small that a data analysis module is totally insufficient. Moreover, the source of data is too subjective in the form of online questionnaires only, and some other supporting data need to be added to enhance the persuasiveness of the argument.

Given the word limit, we aimed to be concise in terms of explaining the methods and data analysis. In relation to the data source comment, we respectfully disagree.  There is a large body of published research that is based on data from online public surveys.  This is a well-established and validated mode of data gathering that does not require alternate data sources to support the survey data.  It is commonly used to source data from widely disperse populations. Given we are focussing on identifying patterns and relationships on perceptions and opinions, there is no “correct” response that requires validation by an additional data source. Examples and critical support for online methods for valid data gathering may be found in:

Babington, A., Hughes, M., Farrell, C., Chambers, J., & Standish, R. J. (2023). Preference for multi-layered, flowering, woody streetscape plantings in a Mediterranean-type climate. Urban Forestry & Urban Greening, 89. https://doi.org/10.1016/j.ufug.2023.128094

Fischer, T. B., Fonseca, A., Geißler, G., Jha-Thakur, U., Retief, F., Alberts, R., & Jiricka-Pürrer, A. (2023). Simplification of environmental and other impact assessments–results from an international online survey. Impact Assessment and Project Appraisal, 41(3), 181-189.

Wang, X., Fielding, K. S., & Dean, A. J. (2023). “Nature is mine/ours”: Measuring individual and collective psychological ownership of nature. Journal of Environmental Psychology, 85, 101919.

Biffignandi, S., & Bethlehem, J. (2021). Handbook of web surveys. John Wiley & Sons.

We would be grateful if some more specific suggestions could be made so we can address the reviewer’s concern and improve the paper.

The presentation of figure 5 is a bit poor, and the distribution icons can be modified, and the corresponding legend, compass, scale and other elements can be added.

We deleted Figure 5 but reinforced Figure 3.

We have also revised the manuscript and fixed grammar errors and clarify some terminology. 

Reviewer 3 Report

Comments and Suggestions for Authors

Today, although there is increasing concern about the urbanization process that leads to the visual and biological homogenization of urban green areas globally, in general, studies that include visual, ecological and social examination of lawns are very limited. The study based on this is valuable and can serve as an example for many similar projects.

However, minor revision is needed.

Author Response

Today, although there is increasing concern about the urbanization process that leads to the visual and biological homogenization of urban green areas globally, in general, studies that include visual, ecological and social examination of lawns are very limited. The study based on this is valuable and can serve as an example for many similar projects.

We thank the reviewer for looking at our manuscript, for the thorough feedback and the appreciation of our research. Our responses to the reviewers’ questions are below.

The information written in lines 493,494,495,496 and even in some similar lines in the article are not included in the findings, it would be more appropriate to give them before the materials and methods.

We followed one of the reviewers’ recommendations and included the sentence in the Introduction to reinforce the importance of the aesthetical function of lawns (lines 74-75:

“One of the most important aesthetical features that make lawns attractive to urban citizens is the green colour and its beauty”. Now our arguments from the survey in the Discussion part about Perth lawns as a colonial heritage and nostalgia for the green colour of the British countryside are better connected to the Introduction part.

It has also been written that a photo survey was actually used to measure participants' preferences for grass alternatives from a series of 5 images. However, it is necessary to write in the method the sources on which this was applied, and such information is not available in the method. For example, if an explanation as follows is added to the article, more satisfactory information will be given to the reader:

Although representing an image with photographs has some limitations (Palmer & Hoffman, 2001, Steinitz, 2001), photosurvey is the most commonly used and valid methodology for the aesthetic evaluation of a landscape that includes a variety of environmental contexts, from cities to agricultural fields to wilderness (Daniel et al. al., 2012; Wang et al., 2016; Tieskens et al., 2018; Aşur & Akpınar Külekçi, 2020; Yazici & Aşur, 2021).

-Aşur, F., & Akpınar Külekçi, E. 2020. The Relationship Between the Adorability of Urban Landscapes and Their Users Demographic Variables: The Case of Edremit, Van/Turkey. Journal of International Environmental Application and Science, 15(1).

-Palmer, J.F. and Hoffman, R.E. 2001. Rating reliability and representation validity in scenic landscape assessments. Landsc. UrbanPlan. 54, 149–161.

-Daniel, T. C., A. Muhar, A. Arnberger, O. Aznar, J. W. Boyd, K. M. Chan, A. and Von Der Dunk. A. 2012. “Contributions of Cultural Services to the Ecosystem Services Agenda.”

Thank you for your valuable suggestion. We included the paragraph in the Method part and also these suggested references. Lines 256-262

“While there are some limitations in using photographs to depict an image to the survey participants [28,29], photo survey remains the most commonly used and reliable methodology for the aesthetic evaluation of a landscape that includes a variety of environmental contexts including urban environment, agricultural fields, and wilderness [30,31,32,33]. The value of using images in the questionnaires for better understanding and visualisation of ecological messages in urban landscapes was also acknowledged by American [34], English [35] and Australian [17] scholars. The most recent example of using direct photographs making in existing demonstration lawn trials in the questionnaire related to residents’ perceptions of and preference for the lawn alternative was done in the city of Xianyang, China [36].”